# RETHINKING RGB COLOR REPRESENTATION FOR IMAGE RESTORATION MODELS

## ABSTRACT

The per-pixel distance loss defined in the RGB color domain has been almost a compulsory choice for training image restoration models, despite its well-known tendency to guide the model to produce blurry, unrealistic textures. To enhance the visual plausibility of restored images, recent methods employ auxiliary objectives such as perceptual or adversarial losses. Nevertheless, they still do not eliminate the reliance on the per-pixel distance in the RGB domain. In this work, we try to redefine the very representation space over which the per-pixel distance is measured. Our augmented RGB ($a$RGB) space is the latent space of an autoencoder that comprises a single affine decoder and a nonlinear encoder, trained to preserve color information while capturing low-level image structures. As a direct consequence, per-pixel distance metrics, $e.g.$, $L_1$, $L_2$, and smooth $L_1$ losses, can also be defined over our $a$RGB space in the same way as for the RGB space. We then *replace* the per-pixel losses in the RGB space with their counterparts in training various image restoration models such as deblurring, denoising, and perceptual super-resolution. By simply redirecting the loss function to act upon the proposed $a$RGB space, we demonstrate boosted performance without any modification to model architectures or other hyperparameters. Our results imply that the RGB color is not the optimal representation for image restoration tasks.

## 1 INTRODUCTION

Since SRCNN (Dong et al., 2016) reinterpreted image restoration pipeline as a cascade of deep neural networks, the field of image restoration has undergone unprecedented improvements, most of which are attributed to the advancements in model architectures (Kim et al., 2016b; Lim et al., 2017; Nah et al., 2017; Tong et al., 2017; Wang et al., 2018b; Zhang et al., 2018b; Waqas Zamir et al., 2021; Liang et al., 2021; Chen et al., 2022). On the contrary, shifting our interest to the very objectives the models are optimized for, we see only a few variations: the per-pixel $L_1$ or $L_2$ distances are used almost unanimously. This particular fondness for the distance metrics *in the RGB color space* stems from the characteristics of the image restoration problem itself, where a low-quality input, the model's reconstruction, and the corresponding ground truth images have extremely dense, pixel-grained correlations in between.

Unfortunately, it is widely known that those per-pixel losses are the main cause of the blurriness easily found in the restored images (Ledig et al., 2017). Each spatial *feature* in the RGB color space is only responsible for the three-dimensional color information at that specific locus; it does not carry any information directly pertaining to local structures. In other words, the models do not learn structural information *from* the loss function. Instead, they only learn it implicitly from its architectural prior. The conventions to remedy the problem are to introduce auxiliary objectives such as perceptual loss (Johnson et al., 2016) or adversarial loss (Ledig et al., 2017; Kupyn et al., 2018; Wang et al., 2018b). Nonetheless, they cannot be used by themselves when accurate reconstruction is required. In particular, a perceptual loss (Johnson et al., 2016) is a distance metric defined over the range of another network, typically a pre-trained classifier (Simonyan and Zisserman, 2015). Those classifiers, despite being favorable latent encoders for the perceptual losses, are originally *designed* to prefer coarse semantic structures over high-frequency textual variations in order to achieve robust classification accuracy. To this end, a classifier typically downscales inputs (Krizhevsky et al., 2012), normalizes internal feature distributions (Ioffe and Szegedy, 2015; Ba et al., 2016), and filters out insignificant patterns using noninvertible rectifiers (von der Malsburg, 1973; Hendrycks and Gimpel,

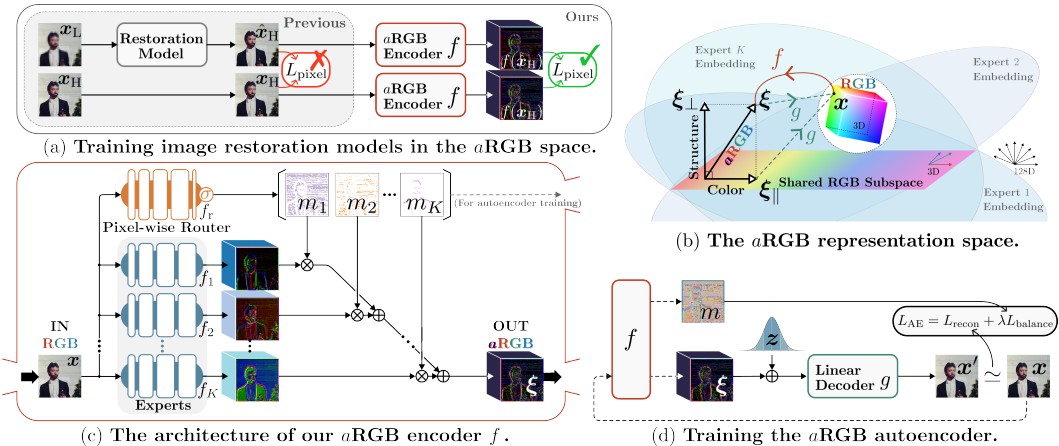

Figure 1: **The proposed $a$RGB representation space.** The augmented RGB ($a$RGB) representation is designed to imbue the gradient-based supervision given by any per-pixel distance metric with rich structural information. Defined with a mixture-of-experts encoder and a linear decoder, our $a$RGB space also gains interpretability, for any $a$RGB encoding is orthogonally decomposable into the color encoding and structure encoding components.

2016). Such process can be advantageous in maintaining semantic information; however, the resulting embeddings inevitably lose information about pixel-grained alignments and colors, which is crucial when we want to reconstruct high-fidelity images that correctly match the given inputs. Adversarial losses (Goodfellow et al., 2014; Ledig et al., 2017; Kupyn et al., 2018; Wang et al., 2018b) cannot be used alone for restoration either, as they prioritize realism over pixel-level accuracy and content preservation. As a consequence, the per-pixel distance metrics have been regarded almost necessary evils in training a restoration network, despite their notoriety of producing blurry outputs.

In summary, yet the per-pixel distances defined over the RGB color representation does provide fine-grained supervision for the paired data, it fails to convey information regarding local structures within an image. On the other hand, despite their structural awareness, existing solutions such as perceptual or adversarial losses cannot change the way of using the per-pixel distances. Because these loss functions do not preserve the exact fine-grained information, the per-pixel distances are still required to assist their supervision. We believe that, however, the lack of structural information within the guidance of per-pixel distances is not attributed to the metrics themselves but rather, to the very space those metrics are defined over, *i.e.*, the RGB color domain. What we need is a representation space where each *pixel* captures its neighboring structure while not losing its original color value so as to provide better supervision with a per-pixel distance. For this goal, we design an encoder that augments images into latent features that satisfy this condition. Our encoder is trained with a *linear* decoder in an autoencoder fashion to ensure those latent features to be decoded back to the original images almost losslessly ($> 60\,\mathrm{dB}$ PSNR). We refer to this latent feature space as the *augmented RGB* ($a$RGB) space. Replacing the RGB representation with our $a$RGB space in calculation of per-pixel distances enjoys several benefits:

**Versatility.** Directly altering the underlying representation space allows us an additional degree of freedom in choosing the loss function. Among various high-performing image restoration models, we choose frameworks employing different per-pixel and auxiliary losses for demonstration, namely: MPRNet (Waqas Zamir et al., 2021), NAFNet (Chen et al., 2022), and ESRGAN (Wang et al., 2018b).

**Performance improvement.** Replacing per-pixel RGB losses with our $a$RGB space-based ones improves not only in perceptual super-resolution tasks but, to our surprise, in the image denoising and deblurring tasks in terms of PSNR and SSIM. Better PSNR metrics could be achieved *without* using the per-pixel RGB distances, despite their mathematical equivalence.

**Interpretability.** In Section 4, we provide comprehensive analysis on our $a$RGB space. Thanks to the *linear* decoder, we can separate the information added to the augmented space from the existing RGB color information. We investigate further into the topology of the $a$RGB space and the characteristics of the gradients from the $a$RGB distances using various visualization techniques.

## 2 LIFTING THE RGB COLOR SPACE

### 2.1 THE *a*RGB AUTOENCODER

Our primary goal is to design a representation space for low-level vision tasks in order to facilitate training of image restoration networks. Designing a representation space is achieved by defining the encoder and the decoder to translate images back and forth between the RGB space and the target space. Building upon the discussion from Section 1, we can split our goal into two parts: (1) the feature at each *pixel* in our space is required to encode its neighboring structure, and (2) the integrity of the color information should be preserved. To fulfill the first requirement, our encoder is a size-preserving ConvNet with nonlinearities to capture the structure among adjacent pixels. For the latter, we employ a per-pixel linear decoder, *i.e.*, a $1 \times 1$ convolution, to strongly constrain the embedding of a pixel to include its RGB color information.

We start from an RGB image $x \in \mathbb{R}^{3 \times H \times W}$. Our convolutional encoder $f$ transforms image $x$ into a feature $\xi \in \mathbb{R}^{C \times H \times W}$ of a new representation space. Unlike typical undercomplete autoencoders, which removes information from its inputs, we aim to *add more* information regarding local structures for each pixel $[\xi]_{ij}$ at coordinate $(i, j)$. Therefore, $C$ must be greater than 3, and the receptive field size $R$ should be greater than unity. Our decoder $g : \xi \mapsto x$ is effectively a single $1 \times 1$ convolution. That is, we can express $g([\xi]_{ij})$ as a per-pixel linear operation: $g([\xi]_{ij}) = A[\xi]_{ij} + b$, where $A \in \mathbb{R}^{3 \times C}$ and $b \in \mathbb{R}^3$. This ensures that each feature $[\xi]_{ij}$ in our representation space extends the color information presented in $[x]_{ij}$, hence the name of our new representation, *augmented* RGB. Additionally, using a linear decoder $g$ offers an interpretability: we can regard the nullspace of $A$, *i.e.*, the set of undecoded information, as a reservoir of any extra information captured by the encoder $f$ other than local colors.

What is crucial at this juncture is to define our *a*RGB space to effectively capture the highly varying, complex mixture of information from the color and the neighboring structure at each pixel. To this end, we employ a mixture-of-experts (MoE) architecture (Jacobs et al., 1991; Shazeer et al., 2017; Fedus et al., 2022) within our encoder. We choose this design based on our conjecture that the topology of the space of image patches is disconnected, and therefore can be more efficiently modeled with a MoE architecture than a single ConvNet. For the set of the smallest images, *i.e.*, a set of pixels, we can argue that their domain is a connected set under absence of quantization, since a pixel can take arbitrary color value. This does not hold in general if the size of the patches become large enough to contain semantic structures. In fact, we cannot interpolate between two images of semantically distinct objects *in the natural image domain*, *e.g.*, there is no such thing as a half-cat half-airplane object *in nature*. This implies that topological disconnectedness emerge from the domain of patches as the size of its patches increases. Since a single-module encoder is a continuous function, learning a mapping over a disconnected set may require deeper architecture with a lot of parameters. An MoE encoder, per contra, can model a discontinuous map more effectively through its discrete routing strategy between small, specialized experts. We will revisit our conjecture in Section 4.

In practice, an RGB image $x \in \mathbb{R}^{3 \times H \times W}$ is fed into the router $f_r$ as well as $K$ encoders $f_1, \ldots, f_K$. The router $f_r$ is a five-layer ConvNet classifier with a softmax at the end. The output of the router $y = f_r(x) \in [0, 1]^{K \times H \times W}$ partitions each pixel of $x$ into $K$ different bins with top-1 policy. This is equivalent to generating mutually exclusive and jointly exhaustive $K$ masks $m_1, \ldots, m_K$ of size $H \times W$. Finally, the features $\xi_1 = f_1(x), \ldots, \xi_K = f_K(\xi)$ are aggregated into a single feature $\xi$, *i.e.*,

$$\xi = f(x) = \sum_{k=1}^{K} m_k \odot f_k(x) = \sum_{k=1}^{K} \mathbb{1}_{\arg\max_{k'}[f_r(x)]_{k'}=k} \odot f_k(x) \in \mathbb{R}^{C \times H \times W}, \quad (1)$$

where $\odot$ is an element-wise multiplication and $\mathbb{1}$ is the indicator function. We ensure that $(g \circ f)(x) = x' \simeq x$ by training $f$ and $g$ jointly in an autoencoder scheme. After the training, the decoder $g$ is discarded and the encoder $f$ is used to generate *a*RGB representations from RGB images.

### 2.2 TRAINING THE AUTOENCODER

Our objective is to ensure that the *a*RGB encoder $f$ effectively learns accurate low-level features from clean (or sharp) and natural images. To achieve this goal, we make use of a dataset $D$, consisting of clean image patches. With this dataset, the *a*RGB autoencoder is trained to minimize the $L_1$ distance between a patch $x \in D$ and its reconstruction $(g \circ f)(x)$. In addition, likewise in Switch Transformer

(Fedus et al., 2022), a load-balancing loss $L_{\text{balance}}$ is applied to encourage the router $f_{\text{r}}$ to distribute pixels evenly across the $K$ experts during training:

$$L_{\text{balance}} = K^2 \sum_{i=1}^{H} \sum_{j=1}^{W} \left[ \max_{k} [f_{\text{r}}(\boldsymbol{x})]_k \right]_{ij}, \tag{2}$$

which is minimized when the distribution is uniform with the value of unity. Furthermore, to increase the sensitivity of the encoder $f$, we simply add an isotropic Gaussian noise at the output of the encoder only during the training of the $a$RGB autoencoder. That is, we have the reconstruction loss:

$$L_{\text{recon}} = \| g(f(\boldsymbol{x}) + \boldsymbol{z}) - \boldsymbol{x} \|_1, \tag{3}$$

where $\boldsymbol{z} \sim \mathcal{N}(\boldsymbol{0}, \boldsymbol{I})$. Although the decoder is only informed with three color channels of each pixel during the training, we observe that the latent space does not degenerate into trivial solutions. See Appendix A for more information. Overall, the training loss for the $a$RGB autoencoder is:

$$L_{\text{AE}} = L_{\text{recon}} + \lambda L_{\text{balance}}. \tag{4}$$

In practice, we choose $\lambda = 0.01$. The final autoencoder achieves 67.21 dB in reconstruction of the Set5 benchmark (Bevilacqua et al., 2012). In other words, the average RGB color difference is below tenth of the quantization step. Henceforth, we will consider our $a$RGB autoencoder lossless in the analysis in Section 4. More implementation details are provided in Appendix B.

## 3 TRAINING IMAGE RESTORATION MODELS IN $a$RGB SPACE

### 3.1 INTEGRATION INTO EXISTING RESTORATION FRAMEWORKS

Training image restoration models with respect to the $a$RGB space only requires a few lines of code modified. An image restoration model is typically trained to minimize a per-pixel distance $L_{\text{pixel}}$, optionally with some auxiliary losses $L_{\text{aux}}$ for perceptual quality, such as a perceptual loss (Johnson et al., 2016) or an adversarial loss (Ledig et al., 2017). The overall loss can be represented as:

$$L_{\text{total}}(\boldsymbol{x}_{\text{H}}, \hat{\boldsymbol{x}}_{\text{H}}) = L_{\text{pixel}}(\boldsymbol{x}_{\text{H}}, \hat{\boldsymbol{x}}_{\text{H}}) + L_{\text{aux}}(\boldsymbol{x}_{\text{H}}, \hat{\boldsymbol{x}}_{\text{H}}), \tag{5}$$

where $\boldsymbol{x}_{\text{H}}$ is the ground-truth image and $\hat{\boldsymbol{x}}_{\text{H}}$ is the restoration result. To train the model in the $a$RGB space, we are only required to modify the the input to the per-pixel loss $L_{\text{pixel}}$. That is, the per-pixel distances are now computed between the images in the $a$RGB space, namely, $f(\boldsymbol{x}_{\text{H}})$ and $f(\hat{\boldsymbol{x}}_{\text{H}})$.

$$L_{\text{total}, \, a\text{RGB}}(\boldsymbol{x}_{\text{H}}, \hat{\boldsymbol{x}}_{\text{H}}) = L_{\text{pixel}}(f(\boldsymbol{x}_{\text{H}}), f(\hat{\boldsymbol{x}}_{\text{H}})) + L_{\text{aux}}(\boldsymbol{x}_{\text{H}}, \hat{\boldsymbol{x}}_{\text{H}}). \tag{6}$$

Since what we present is not a specific loss function but the underlying *space* itself, our method can be seamlessly integrated with any existing restoration framework regardless of the type of per-pixel loss it uses. Typical per-pixel losses used for these tasks can be grouped into three categories: an $L_1$ loss; an $L_2$ loss and its equivalents; and a group of smooth $L_1$ losses that interpolate between the former two. To demonstrate the versatility of our solution, we choose a high-performing image restoration model trained by a loss from each of the group to solve different type of tasks. In specific, a perceptual image super-resolution model trained for an $L_1$ loss, a real image denoising model trained for a PSNR loss, an equivalent to the $L_2$ loss, and finally a motion blur deblurring model trained for a Charbonnier loss, a type of smooth $L_1$ loss, are chosen. A notable feature of our method is that the trained image restoration models with respect to our $a$RGB representation space are generally better at reconstructing the underlying edge structures. This offers visual artifact reduction for perceptual image super-resolution in Section 3.2, sharper edges and enhanced alignments for image denoising and deblurring in Section 3.3 and 3.4. More visual comparisons are provided in Appendix D.

### 3.2 PERCEPTUAL IMAGE SUPER-RESOLUTION WITH $L_1$ LOSS

Our initial hypothesis revolved around the potential of our $a$RGB encoder $f$ to enrich the supervision of the per-pixel loss with structural information. Perceptual super-resolution should be a natural starting point to search for the evidence, since in the task, the supervision from the original per-pixel loss is heavily interfered by structure-aware auxiliary losses, *i.e.*, the VGG perceptual loss (Simonyan and Zisserman, 2015; Johnson et al., 2016) and the adversarial loss (Ledig et al., 2017). We trained ESRGAN (Wang et al., 2018b) models and summarized the results in Table 1. Fine-tuned over the

Table 1: **Quantitative results on training** $4\times$ **super-resolution ESRGAN in the** $a$RGB **space.** In our methods using $a$RGB representation, we modify only the $L_1$ loss by exchanging it with the $L_{1,aRGB}$ loss. All the other training hyperparameters are left untouched. Better scores in each block are shown in **boldface** text.

| | DIV2K-Val | | | | | Urban100 | | | | |
|---|---|---|---|---|---|---|---|---|---|---|
| Objective | PSNR↑ | SSIM↑ | LPIPS↓ | NIQE↓ | FID↓ | PSNR↑ | SSIM↑ | LPIPS↓ | NIQE↓ | FID↓ |
| Pre-trained RRDBNet[†] | 29.466 | 0.8306 | 0.2537 | 5.4860 | 15.910 | 25.496 | 0.7951 | 0.1963 | 5.6236 | 23.729 |
| $0.01L_1$         $+0.005L_{\text{Adv}}$ | 27.102 | **0.7687** | 0.1282 | **3.0419** | 13.593 | **23.535** | **0.7373** | 0.1322 | 3.9479 | 18.428 |
| $0.01L_{1,aRGB}$    $+0.005L_{\text{Adv}}$ | **27.218** | 0.7622 | **0.1235** | 3.0896 | **12.936** | 23.348 | 0.7204 | **0.1289** | **3.8524** | **18.015** |
| $0.01L_1$   $+L_{\text{VGG}}+0.005L_{\text{Adv}}$[†] | 26.627 | 0.7033 | 0.1154 | 3.0913 | 13.557 | 22.776 | 0.7033 | 0.1232 | 4.2067 | 20.616 |
| $0.01L_{1,aRGB}+L_{\text{VGG}}+0.005L_{\text{Adv}}$ | **26.845** | **0.7500** | **0.1110** | **2.9615** | **12.799** | **23.270** | **0.7196** | **0.1183** | **3.8982** | **17.739** |

[†]The official ESRGAN model (Wang et al., 2018b).

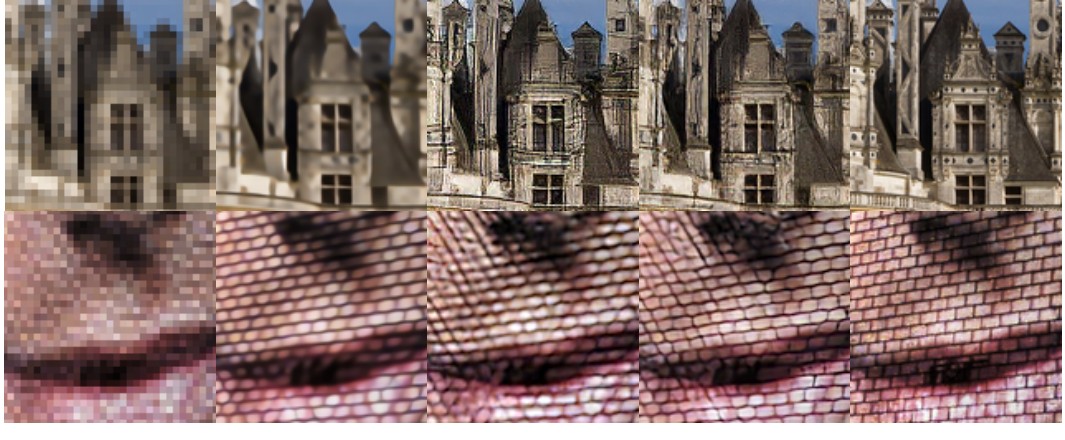

     (a) LR          (b) RRDBNet       (c) ESRGAN      (d) $L_{1,aRGB}+L_V+L_A$     (e) HR

Figure 2: **Qualitative comparison of ESRGAN models trained with different loss functions.** Each column corresponds to each row in Table 1. The loss weights are omitted for brevity, ESRGAN corresponds to the $0.01L_1+L_{\text{VGG}}+0.005L_{\text{Adv}}$ in Table 1.

same PSNR-oriented pre-trained RRDBNet, various combinations for the adversarial training are examined. Here, our method simply modifies the $L_1$ loss to act within the $a$RGB space.

First, as Table 1 indicates, the modified $L_1$ metric, $L_{1,aRGB}$, provides sufficient constraints for stabilizing the adversarial training of a super-resolution model. Remarkably, even in the absence of the perceptual loss, our $L_{1,aRGB}$ loss generally improves perceptual scores over the original $L_1$ loss while maintaining similar PSNR scores during adversarial training. This implies that our $a$RGB representation provides complementary information that the conventional per-pixel $L_1$ distances does not provide. Furthermore, the last two rows of Table 1 demonstrate that the benefit of training in our $a$RGB space is maximized in the presence of the perceptual loss. This implies that the local structural information captured within our $a$RGB representation is also complementary to the supervision from a pre-trained classifier. As a result, this leads to superior performance in every distortion-based and perceptual metric compared to the original ESRGAN. In particular, the improvements in the PSNR and SSIM scores aligns with our design philosophy that the RGB colors are *included* as a subspace in our $a$RGB representation; in other words, the effect of minimizing the $L_1$ loss can also be achieved by minimizing the $L_{1,aRGB}$ loss. From visual results in Figure 2 and Appendix D, we can observe how artifacts are suppressed using our $L_{1,aRGB}$ loss, successfully guiding the adversarial training towards visually pleasing restoration. More quantitative results are provided in Appendix C.

### 3.3    REAL NOISE DENOISING WITH $L_2$ LOSS

To demonstrate the effect of $a$RGB representation with $L_2$ loss, we choose NAFNet (Chen et al., 2022), which employs a per-pixel PSNR loss $L_{\text{PSNR}}$, a mathematically equivalent form of the $L_2$ loss. We first train a NAFNet-width32 on the SIDD Medium sRGB dataset (Abdelhamed et al., 2018) with our new PSNR loss $L_{\text{PSNR},aRGB}$, the same metric but defined within the $a$RGB space. To our surprise, Table 2 and Figure 3 reveal that our $a$RGB representation provides better PSNR and SSIM scores than the original model directly trained using the PSNR metric $L_{\text{PSNR}}$. The results imply that our $a$RGB representation not only maintains most of original RGB information but also incorporates additional local structural information that leads to better supervision in the denoising task. Additional experiments using different metrics for the same task reveal another noteworthy characteristics of

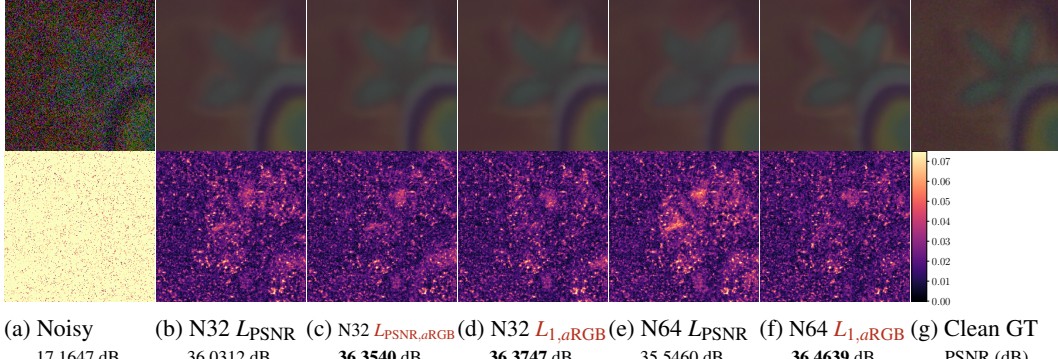

(a) Noisy    (b) N32 $L_{\text{PSNR}}$   (c) N32 $L_{\text{PSNR},a\text{RGB}}$ (d) N32 $L_{1,a\text{RGB}}$(e) N64 $L_{\text{PSNR}}$   (f) N64 $L_{1,a\text{RGB}}$ (g) Clean GT

17.1647 dB    36.0312 dB    **36.3540** dB    **36.3747** dB    35.5460 dB    **36.4639** dB    PSNR (dB)

Figure 3: **Qualitative comparison of real image denoising models trained with different loss functions.** Each column corresponds to each row in Table 2. N32 corresponds to NAFNet-width32 and N64 corresponds to NAFNet-width64. The bottom row shows the maximum absolute difference in color with a range of $[0, 1]$.

changing the representation space. As elaborated in Section 4.3, changing the underlying space can profoundly alters the scale and the shape of a metric and its gradients, resulting in different training dynamics. A direct consequence is that the optimal hyperparameters and their resulting performance may change for restoration framework in use. Better performance obtained with NAFNets trained for the $L_1$ metric in our $a$RGB space in the last rows of Table 3 clearly demonstrates this issue, revealing a potential unexpected benefit from changing the underlying representation.

### 3.4 MOTION BLUR DEBLURRING WITH SMOOTH $L_1$ LOSS

A Charbonnier loss (Bruhn et al., 2005) is a type of smooth $L_1$ loss defined as $L_{\text{Char}}(\hat{x}_{\text{H}}, x_{\text{H}}) = (\|\hat{x}_{\text{H}} - x_{\text{H}}\|_2^2 - \varepsilon^2)^{1/2}$, where $\varepsilon$ is a small constant. To show the effectiveness of our $a$RGB representation with this type of loss, we train an MPRNet (Waqas Zamir et al., 2021) for motion blur deblurring task using GoPro dataset (Nah et al., 2017). The MPRNet is originally trained with a Charbonnier loss with $\varepsilon = 10^{-3}$ together with an edge loss, an auxiliary loss defined as another Charbonnier loss calculated between the Laplacians of two images. We leave the edge loss and its weight untouched and change only the Charbonnier loss to act upon our $a$RGB space, *i.e.*, $L_{\text{MPRNet}, a\text{RGB}} = L_{\text{Char}}(f(\hat{x}_{\text{H}}), f(x_{\text{H}})) + 0.05 L_{\text{Char}}(\Delta \hat{x}_{\text{H}}, \Delta x_{\text{H}})$. We observe clear improvements in Table 3 and Figure 4. As shown, the performance gain was orthogonal to existing enhancement techniques, *e.g.*, test-time local converter (TLC) (Chu et al., 2022). From the experiments, we conclude that our $a$RGB representation indeed helps training image restoration models better than the RGB color representation in a variety of tasks, architectures, loss functions, and lead to synergic effect with a variety of other enhancement techniques, such as perceptual loss, adversarial training, edge loss, and test-time local converter.

## 4 DISCUSSION

In order to understand the representation learned by the $a$RGB autoencoder, we first explore the consequence of our two key design choices: the *linear* decoder and the *mixture-of-experts* encoder.

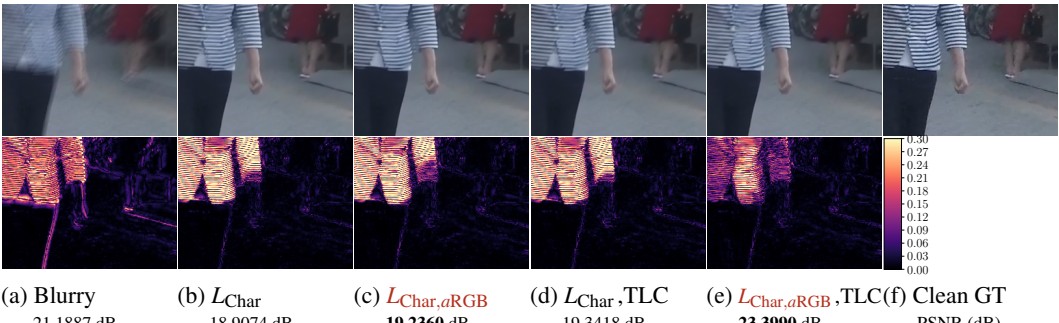

(a) Blurry    (b) $L_{\text{Char}}$    (c) $L_{\text{Char},a\text{RGB}}$    (d) $L_{\text{Char}}$,TLC    (e) $L_{\text{Char},a\text{RGB}}$,TLC(f) Clean GT

21.1887 dB    18.9074 dB    **19.2360** dB    19.3418 dB    **23.3990** dB    PSNR (dB)

Figure 4: **Qualitative comparison of motion blur deblurring models trained with different loss functions.** Each column corresponds to each row in Table 3. The bottom row is the maximum absolute RGB difference.

Table 2: **Results on real image denoising using NAFNet.**

| | | SIDD | |
|---|---|---|---|
| Model | Objective | PSNR↑ | SSIM↑ |
| NAFNet-width32 | $L_{\text{PSNR}}$ | 39.9672 | 0.9599 |
| NAFNet-width32 | $L_{\text{PSNR},a\text{RGB}}$ | **39.9864** | **0.9601** |
| NAFNet-width32 | $L_{1,a\text{RGB}}$ | **40.0106** | **0.9602** |
| NAFNet-width64 | $L_{\text{PSNR}}$ | 40.3045 | 0.9614 |
| NAFNet-width64 | $L_{1,a\text{RGB}}$ | **40.3364** | **0.9620** |

Table 3: **Results on motion blur deblurring using MPRNet.**

| | | GoPro | | HIDE | |
|---|---|---|---|---|---|
| Model | Objective | PSNR↑ | SSIM↑ | PSNR↑ | SSIM↑ |
| MPRNet | $L_{\text{Char}} + 0.05 L_{\text{Edge}}$ | 32.6581 | 0.9589 | 30.9622 | 0.9394 |
| MPRNet | $L_{\text{Char},a\text{RGB}} + 0.05 L_{\text{Edge}}$ | **32.7118** | **0.9594** | **31.0248** | **0.9398** |
| MPRNet-TLC | $L_{\text{Char}} + 0.05 L_{\text{Edge}}$ | 33.3137 | 0.9637 | 31.1868 | 0.9418 |
| MPRNet-TLC | $L_{\text{Char},a\text{RGB}} + 0.05 L_{\text{Edge}}$ | **33.3886** | **0.9642** | **31.2082** | **0.9421** |

(a) Inverting orthogonal mixture of two $a$RGB embeddings. (b) Expert selection map of the MoE router $f_r$. (c) t-SNE plot of the $a$RGB embedding $\boldsymbol{\xi}$ of pixels in image 5b. (d) Change of $L_2$ metrics in the $a$RGB space relative to the $L_2$ metrics in the RGB space.

Figure 5: **Understanding the learned $a$RGB representation.** Figure 5a show a visual example of $a$RGB embedding inversion. Figure 5b and 5c reveal clear evidence that the experts of our $a$RGB encoder $f$ are specialized for a particular type of input structures, and that even the embedding vectors within a single patch are clustered in a complicated manner, justifying our usage of MoE architecture. Figure 5d shows how the distance metric changes in the $a$RGB space relative to the distance in the RGB space. Mean distances and their standard deviations are measured by MSE losses between an image and the same image with 100 AWGNs with the same standard deviation. Note that the $a$RGB space slightly exaggerates the distance more outside natural image domain, *e.g.*, Gaussian noise, and the metric's variance is negligibly small.

Then, we quantify the effect of changing the representation space on the scale of metrics defined over the space and their gradients. We conclude our discussion with ablation studies.

## 4.1 NULLSPACE OF THE DECODER

In addition to the design simplicity, our pixel-wise *linear* decoder enjoys an additional benefit: decomposability. Since our autoencoder is almost lossless as demonstrated in Table 8, we will consider that the RGB $\boldsymbol{x} \in \mathbb{R}^3$ and the $a$RGB $\boldsymbol{\xi} = f(\boldsymbol{x}) \in \mathbb{R}^C$ representations of any given image equivalent. That is, $\boldsymbol{x}' = g(\boldsymbol{\xi}) = \boldsymbol{A}\boldsymbol{\xi} + \boldsymbol{b} = \boldsymbol{x}$. As a result of the linearity of our decoder $g$, the $a$RGB representation $\boldsymbol{\xi}$ can be decomposed into the sum of two orthogonal components:

$$\boldsymbol{\xi} = \boldsymbol{\xi}_{\parallel} + \boldsymbol{\xi}_{\perp}, \quad \text{s.t.} \quad \boldsymbol{\xi}_{\parallel} = \boldsymbol{A}^{\dagger}\boldsymbol{A}\boldsymbol{\xi} =: f_{\parallel}(\boldsymbol{x}) \quad \text{and} \quad \boldsymbol{\xi}_{\perp} = (\boldsymbol{I} - \boldsymbol{A}^{\dagger}\boldsymbol{A})\boldsymbol{\xi} =: f_{\perp}(\boldsymbol{x}), \quad (7)$$

where $\boldsymbol{A}^{\dagger}$ is the Moore-Penrose pseudoinverse of $\boldsymbol{A}$. The parallel component $\boldsymbol{\xi}_{\parallel}$ of the $a$RGB representation lies in the three-dimensional subspace of $\mathbb{R}^C$ that is projected onto the RGB colors by the decoder $g$, *i.e.*, $\boldsymbol{A}\boldsymbol{\xi}_{\parallel} = \boldsymbol{A}\boldsymbol{A}^{\dagger}\boldsymbol{A}\boldsymbol{\xi} = \boldsymbol{A}\boldsymbol{\xi}$. The remaining perpendicular part $\boldsymbol{\xi}_{\perp}$ can be regarded as the information the $a$RGB space encodes in addition to the RGB colors. The contribution of the two components can be visualized by inverting the encoder $f$ with respect to a mixed embedding:

$$f^{-1}(\boldsymbol{\xi}_{\text{mix}}) = \arg\min_{\boldsymbol{z}} \|f(\boldsymbol{z}) - \boldsymbol{\xi}_{\text{mix}}\|_2^2, \text{ s.t. } \boldsymbol{\xi}_{\text{mix}} = \boldsymbol{\xi}_{1\parallel} + \boldsymbol{\xi}_{2\perp} = \boldsymbol{A}^{\dagger}\boldsymbol{A}f(\boldsymbol{x}_1) + (\boldsymbol{I} - \boldsymbol{A}^{\dagger}\boldsymbol{A})f(\boldsymbol{x}_2). \quad (8)$$

We use a SGD optimizer with a learning rate of 0.1 for 50 iterations. As shown in Figure 5a and Appendix E, the inversion of the mixed embedding inherits color information from the parallel embedding $\boldsymbol{\xi}_{1\parallel}$, while the perpendicular part $\boldsymbol{\xi}_{2\perp}$ contributes to the high-frequency edge information.

## 4.2 SPECIALIZATION OF THE EXPERTS AND LEARNED STRUCTURES

Figure 5b visualizes how individual pixels of a natural image are distributed into $K = 20$ experts. Unlike in semantic segmentation, where segmentation maps are chunked into large blocks of semantically correlated pixels, our pixel-wise router $f_r$ generates fine-grained distributions of pixels. That is, multiple experts jointly involve in encoding the same texture such as the blue sky and the leafy trees. Another salient feature we can observe in the figure is that edges of different orientations are dealt with different experts, implying their specialization. Visualizing the $a$RGB embedding space using t-SNE (van der Maaten and Hinton, 2008) provides us with additional insights on the topology of

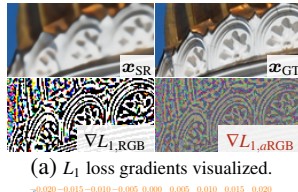

(a) $L_1$ loss gradients visualized.

(b) Histogram of gradients of 6a.

Figure 6: **Gradients from** $L_{1,\mathbf{RGB}}$ **and** $L_{1,\mathbf{aRGB}}$ **losses.**

Table 4: **Ablation studies on the** *a*RGB **autoencoder.** RRDBNets (Wang et al., 2018b) are trained with DIV2K (Agustsson and Timofte, 2017) for 300k iterations for $4\times$ SISR tasks with only the $L_1$ loss between the *a*RGB embeddings.

| RRDBNet in $4\times$ SISR | | | | Set14 | | Urban100 | | DIV2K-Val | |
|---|---|---|---|---|---|---|---|---|---|
| # experts | Routing | *a*RGB train set | Reg. noise | PSNR↑ | SSIM↑ | PSNR↑ | SSIM↑ | PSNR↑ | SSIM↑ |
| 1 | - | DIV2K | ✓ | 26.87 | 0.7467 | 24.75 | 0.7735 | 29.08 | 0.8222 |
| 5 | MoE | DIV2K | ✓ | 26.87 | 0.7477 | 24.83 | 0.7745 | 29.12 | 0.8231 |
| 10 | MoE | DIV2K | ✓ | 26.89 | 0.7474 | 24.84 | 0.7750 | 29.11 | 0.8231 |
| **20** | MoE | DIV2K | ✓ | 26.91 | 0.7471 | 24.87 | 0.7745 | 29.14 | 0.8227 |
| 30 | MoE | DIV2K | ✓ | 26.89 | 0.7476 | 24.84 | 0.7750 | 29.11 | 0.8231 |
| 20 | MoE | GoPro | ✓ | 26.89 | 0.7459 | 24.83 | 0.7728 | 29.12 | 0.8220 |
| 20 | MoE | SIDD | ✓ | 26.86 | 0.7420 | 24.80 | 0.7691 | 29.08 | 0.8186 |
| 20 | MoE | Noise | ✓ | 26.65 | 0.7461 | 24.64 | 0.7729 | 28.87 | 0.8212 |
| 20 | MoE | DIV2K | ✗ | 26.91 | 0.7469 | 24.85 | 0.7722 | 29.13 | 0.8223 |

the space. Figure 5c reveals that the *a*RGB embeddings cluster into multiple disconnected groups in two different types: *common* groups where multiple experts are involved in encoding process and *specialized* groups where a single expert is exclusively allocated for the embeddings. These observations align well with our initial design principles in Section 2.1, where the feature embeddings occupy highly complicated, disconnected set, and an MoE architecture effectively deals with this structure by specializing each expert to a subset of the embedding space.

### 4.3 *a*RGB METRIC SPACE AND PRODUCED GRADIENTS

The main purpose of our the *a*RGB space is to provide alternative supervision to the existing image restoration framework. This supervision is realized with a metric defined over the space and its gradients generated from pairs of images. To this end, we first visualize the correlation between $L_2$ distances defined in the RGB and our *a*RGB spaces in Figure 5d. We plotted additional figure with title $X - \overline{X_0 X_1}$ to show the deviation of the graph over the straight line, showing clear convexity of the graph. This implies that the metrics within *a*RGB spaces are inflated when the given two images are similar. Figure 6 shows the gradients from two per-pixel $L_1$ losses between a restored image and its high-quality counterpart defined over both spaces. Unlike RGB $L_1$ loss which exhibits a highly off-centered, discrete distribution, the $L_{1,a\mathrm{RGB}}$ loss shows smooth and centered distribution of gradients. We believe that this allows for the stable training of the image restoration models despite its huge scale of the generated gradients from the $L_{1,a\mathrm{RGB}}$ loss, which is more than a hundredfold as shown in the x axis of Figure 6b. In the RGB domain, the same scale of gradient is achievable only through increasing the learning rate, which leads to destabilization of the training. Overall, the analyses show how our *a*RGB encoder helps the training of image restoration models.

### 4.4 ABLATION STUDY

Lastly, we provide ablation studies to determine the best hyperparameters for our *a*RGB autoencoder. We compare the models by the results of training an RRDBNet (Wang et al., 2018b) only on DIV2K dataset. The results are summarized in Table 4. More information is elaborated in Appendix B.

**Number of experts.** The first four rows of Table 4 show the effect of the number of experts of the *a*RGB encoder $f$ on its supervision quality. From the results, we choose to fix the number of experts to 20 throughout our experiments.

**Dataset dependence.** As the second part of Table 4 presents, the training data for the *a*RGB autoencoder decides the quality of supervision the model gives. This implies that our *a*RGB autoencoder utilizes structural priors of its training data. Appendix 7 provides additional theoretical and empirical evidence that our *a*RGB autoencoder learns image structures to reconstruct given images.

**Regularizers.** In the last row of Table 4, we observe that the regularizing noise $z$ added at the end of the encoder during training helps the *a*RGB encoder to produce stronger supervision for image restoration models. In practice, we observe more than tenfold reduction in the scale of produced gradients when the *a*RGB autoencoder trained without the regularizing noise is applied. This correlates to our discussion in Section 4.3, that our *a*RGB encoder helps training image restoration models by stably increasing the scale of gradients.

## 5 RELATED WORK

**Pairwise loss in image restoration.** Training a deep neural network that translates low-quality images into high-quality estimates has undoubtedly become the standard way of solving image restoration. While most of the advancements have been made in the network architecture (Kim et al., 2016b; Lim et al., 2017; Nah et al., 2017; Tong et al., 2017; Wang et al., 2018b; Zhang et al., 2018b; Waqas Zamir et al., 2021; Liang et al., 2021; Waqas Zamir et al., 2022; Chen et al., 2022), the importance of loss functions is also widely acknowledged. Since SRCNN (Dong et al., 2016), the first pioneer, employed the MSE loss, the first image restoration models had been trained with the MSE loss (Kim et al., 2016a;b; Nah et al., 2017; Zhang et al., 2017). However, after EDSR (Lim et al., 2017) reported that better convergence can be achieved with $L_1$ loss, various pairwise loss functions are explored. LapSRN (Lai et al., 2017) rediscovers Charbonnier loss (Bruhn et al., 2005), a type of smooth $L_1$ loss, for image super-resolution, which is also employed in image deraining (Jiang et al., 2020) with a new edge loss, defined as a Charbonnier loss between Laplacians, which is then employed in general restoration by MPRNet (Waqas Zamir et al., 2021). NAFNet (Chen et al., 2022), on the other hand, uses the PSNR score directly as a loss function. In accordance with these approaches, we attempt a more general approach to design a representation space, over which those loss functions can be redefined.

**Structural prior of natural images.** It is generally recognized that a convolutional neural network, either trained (Simonyan and Zisserman, 2015) or even untrained (Ulyanov et al., 2018), contains structural prior that resonates with the internal structure of natural images. This prior information permeates through the network into its output space. Attempts to exploit this information include the perceptual loss (Johnson et al., 2016) and various perceptual metrics (Zhang et al., 2018a; Ding et al., 2020). Those are pairwise distance metrics defined over the range space of pre-trained classifier networks (Krizhevsky et al., 2012; Simonyan and Zisserman, 2015). However, as mentioned in Section 1, such losses cannot be used alone when it is required to respect the strong correspondence between the generated and the desired images. Different from the strategies sought for the perceptual metrics, our *a*RGB encoder is designed to preserve the full information of its inputs by a scale-preserving architecture and a linear decoder to strictly constrain the representation.

**Mixture of Experts.** Instead of relying on a single model to handle complex large-scaled data, a more effective approach is to distribute the workload among multiple workers. To achieve this, a routing strategy (Shazeer et al., 2017) can be employed to divide information between different models, each of which processes a subset of the training data. These individual models, referred to as *experts*, collectively form a Mixture of Experts (MoE) (Jacobs et al., 1991). Recent studies (Zhou et al., 2022; Fedus et al., 2022) have shown the advantages of MoE in deep learning. However, there are two main challenges when working with multiple experts: limited computational resources and training stability. The conventional routing strategy can lead to unstable training of the MoE unless appropriate regularization methods are applied. Moreover, without advanced techniques (Fedus et al., 2021; He et al., 2021), MoE experience longer processing times as the number of experts increases. In response to these challenges, we employ a balancing loss (Fedus et al., 2022) to ensure the stable training of expert networks and incorporate MoE exclusively during the training phase, leaving the testing phase unaffected.

## 6 CONCLUSION

It is a well-known phenomenon (Ledig et al., 2017) that per-pixel pairwise loss functions, such as $L_1$ or $L_2$ distances, defined in the RGB color space have a strong tendency to guide the trained image restoration model to produce blurry, unrealistic textures. We hypothesize that such problem can be alleviated if we have a representation space that contains accurate color information as well as the local structural information of an image. Our augmented RGB (*a*RGB) representation is designed with a nonlinear mixture-of-experts encoder and a linear decoder to meet the requirements. From diversified set of experiments, we demonstrate the improved performance across a variety of image restoration tasks such as perceptual super-resolution, denoising, and deblurring could be achieved by only changing the representation space to our *a*RGB space. Given our results suggesting that the RGB color space may not be the optimal representation space for low-level computer vision tasks, we hope our work spurs more interests and exploration in this research direction.

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

# A THEORETICAL STUDY ON THE STRUCTURE EMBEDDED IN $a$RGB SPACE

The goal of this section is to provide a simple theoretical analysis on how the structure is learned in our $a$RGB encoder $f$. This section comprises of two parts. The first part shows that our $a$RGB encoder is a piecewise linear function over the connected neighborhood regions in the RGB pixel domain. From this, we can equivalently transform our $a$RGB autoencoder to a coordinate-wise function, which is useful in the upcoming analyses. In the second part, we will show that an autoencoder with a neural network encoder and a linear decoder like ours does not learn structural priors if it is perfectly lossless. In other words, we claim that imperfect, yet almost perfect ($> 60$dB PSNR), reconstruction capability of our $a$RGB autoencoder helps its encoder to learn local structures within natural images. Based on the mathematical analysis, we provide another method to measure how much image structure is captured in the $a$RGB autoencoder.

## A.1 THE $a$RGB ENCODER IS A PIECEWISE LINEAR FUNCTION OVER THE LOCAL NEIGHBORHOOD STRUCTURES

It is known that a multi-layer perceptron (MLP) with a continuous piecewise linear activation is also a continuous piecewise linear (CPWL) function (Rahaman et al., 2019). That is, given an input $\boldsymbol{x} \in \mathbb{R}^C$, the network $f$ has an output $\boldsymbol{y} = f(\boldsymbol{x}) \in \mathbb{R}^{C'}$, which can be explicitly written as:

$$f(\boldsymbol{x}) = \sum_{\varepsilon} \mathbb{1}_{P_\varepsilon(\boldsymbol{x})}(\boldsymbol{W}_\varepsilon \boldsymbol{x} + \boldsymbol{b}_\varepsilon), \tag{9}$$

where $\varepsilon$ is an index of a (connected) region $P_\varepsilon \subset \mathbb{R}^C$ and $\mathbb{1}_{P_\varepsilon}$, the region's indicator function. $\boldsymbol{W}_\varepsilon \in \mathbb{R}^{C' \times C}$ and $\boldsymbol{b}_\varepsilon \in \mathbb{R}^{C'}$ are the effective weight and bias at the region $P_\varepsilon$, respectively. This interpretation of neural networks is straightforward for multilayer perceptrons. An MLP is a composition of linear layers and rectifiers, which are indeed continuous and piecewise linear, and a composition of a continuous piecewise linear function is also continuous and piecewise linear.

Convolutional neural networks with rectifiers are no different. A linear convolution operation

$$[\boldsymbol{k} \star \boldsymbol{x}]_{h,w} := \sum_{i=-\lfloor k_h/2 \rfloor}^{\lfloor k_h/2 \rfloor} \sum_{j=-\lfloor k_w/2 \rfloor}^{\lfloor k_w/2 \rfloor} [\boldsymbol{k}]_{i,j}[\boldsymbol{x}]_{h+i,w+j}, \tag{10}$$

where $k_h$ and $k_w$ define the height and width of the discrete kernel, can be viewed as a coordinate-wise linear layer with a flattened weight operating on a concatenation of translations of the input $\boldsymbol{x}$, *i.e.*,

$$\boldsymbol{k} \star \boldsymbol{x} = \boldsymbol{W}_{\boldsymbol{k}} \tilde{\boldsymbol{x}}, \quad \text{where} \tag{11}$$
$$\boldsymbol{W}_{\boldsymbol{k}} = \text{flatten}(\boldsymbol{k}), \quad \text{and} \tag{12}$$
$$\tilde{\boldsymbol{x}} = \underset{i,j}{\text{concat}}[\text{translate}[\boldsymbol{x},(i,j)]]. \tag{13}$$

In other words, with a receptive field of size $R \times R$, an image $x \in \mathbb{R}^{C \times H \times W}$ is (linearly) transformed into the extended space $\tilde{x} \in \mathbb{R}^{CR^2 \times H \times W}$, and the ConvNet $f : \mathbb{R}^{C \times H \times W} \to \mathbb{R}^{C' \times H \times W}$ is equivalent to the coordinate-wise function $\tilde{f} : \mathbb{R}^{CR^2} \to \mathbb{R}^{C'}$, which is continuous and piecewise linear.

Since our transform from $f$ to $\tilde{f}$ can be applied to all the $K$ experts $f_1, \ldots f_K$ of our $a$RGB autoencoder, we can abstract away the coordinates for simplicity. Another virtue of this reform is that we do not need to care about the router $f_r$ in our analysis hereafter. From equation 1, for each coordinate $c \in [H] \times [W]$, we have:

$$[\boldsymbol{\xi}]_c = [f(\boldsymbol{x})]_c = \sum_{k=1}^{K} [m_k \odot f_k(\boldsymbol{x})]_c = \sum_{k=1}^{K} [m_k]_c [f_k(\boldsymbol{x})]_c = \sum_{k=1}^{K} [m_k]_c \tilde{f}_k([\tilde{\boldsymbol{x}}]_c) =: \tilde{f}([\tilde{\boldsymbol{x}}]_c), \tag{14}$$

and since the coordinate-wise equivalent of each expert $\tilde{f}_k$ is continuous and piecewise linear, its weighted summation is also piecewise linear, yet it may not be continuous. Moreover, the function's argument $[\tilde{\boldsymbol{x}}]_c \in \mathbb{R}^{CR^2}$ is a reshaping of the receptive field of size $R \times R$ at the locus $c$. As a result, our mixture of experts encoder $f$ is equivalent to a piecewise linear, yet not generally continuous, function over $R \times R$ neighborhood of each pixel in the RGB representation $\boldsymbol{x}$. No matter how each pixel is distributed by the router, each feature vector $[\boldsymbol{\xi}]_c$ at an arbitrary coordinate $c \in [H] \times [W]$ is a piecewise linear function of a local $R \times R$ neighborhood of image $\boldsymbol{x}$ at the locus $c$.

## A.2 A FLAWLESS AUTOENCODER IS A BAD STRUCTURE ENCODER

From the previous analysis, we may omit the spatial coordinate $c$ and assume $\boldsymbol{x} \in \mathbb{R}^C$ be an image in the RGB color space, $\boldsymbol{\xi} \in \mathbb{R}^{C'}$ be the same image in the $a$RGB representation. Let $\tilde{\boldsymbol{x}} = [\boldsymbol{x} \quad B(\boldsymbol{x})]^\top \in \mathbb{R}^{CR^2}$ be the flattened $R \times R$ neighborhood patch in the RGB domain. $B : \mathbb{R}^C \to \mathbb{R}^{C(R^2-1)}$ is a structure function that maps a center pixel $\boldsymbol{x}$ to its peripheral pixels. For the simplicity, we discard the function notation and regard $B$ as a vector in $\mathbb{R}^{C(R^2-1)}$ from now on. The $a$RGB autoencoder is equivalently reformed coordinate-wise. Let our $a$RGB encoder be $f : \mathbb{R}^{CR^2} \to \mathbb{R}^{C'}$ and the $a$RGB decoder be $g : \mathbb{R}^{C'} \to \mathbb{R}^C$. Note that we discard tilde from the encoder equivalent $\tilde{f}$ for the sake of simplicity. Again, since the decoder is linear, we can also write $g(\boldsymbol{\xi}) = \boldsymbol{A}\boldsymbol{\xi} + \boldsymbol{b}$, where $\boldsymbol{A} \in \mathbb{R}^{C \times C'}$ and $\boldsymbol{b} \in \mathbb{R}^C$.

The piecewise linear characteristic of the encoder $f$ lets us rewrite the autoencoder into a form:

$$f(\tilde{\boldsymbol{x}}) = \sum_\varepsilon \mathbb{1}_{P_\varepsilon(\tilde{\boldsymbol{x}})}(\boldsymbol{W}_\varepsilon \tilde{\boldsymbol{x}} + \boldsymbol{b}_\varepsilon) = \boldsymbol{\xi}, \tag{15}$$

$$g(\boldsymbol{\xi}) = \boldsymbol{A}\boldsymbol{\xi} + \boldsymbol{b} = \boldsymbol{x}' \simeq \boldsymbol{x}, \tag{16}$$

following equation 9. The autoencoder $h = g \circ f$ can be written as:

$$h(\tilde{\boldsymbol{x}}) = g(f(\tilde{\boldsymbol{x}})) = \boldsymbol{A}\sum_\varepsilon \mathbb{1}_{P_\varepsilon(\tilde{\boldsymbol{x}})}(\boldsymbol{W}_\varepsilon \tilde{\boldsymbol{x}} + \boldsymbol{b}_\varepsilon) + \boldsymbol{b}, \tag{17}$$

$$= \sum_\varepsilon \mathbb{1}_{P_\varepsilon(\tilde{\boldsymbol{x}})} \left(\boldsymbol{A}\boldsymbol{W}_\varepsilon \tilde{\boldsymbol{x}} + \boldsymbol{A}\boldsymbol{b}_\varepsilon + \boldsymbol{b}\right). \tag{18}$$

Let $\varepsilon'$ is a subscript for a connected region in the partition $P$ that contains the coordinate $c$. That is, $\mathbb{1}_{P_{\varepsilon'}(\tilde{\boldsymbol{x}})} = 1$ and $\mathbb{1}_{P_\varepsilon(\tilde{\boldsymbol{x}})} = 0$ for $\varepsilon \neq \varepsilon'$. The summation from equation 18, then, can be simplified:

$$h(\tilde{\boldsymbol{x}}) = \boldsymbol{A}\boldsymbol{W}_{\varepsilon'}\tilde{\boldsymbol{x}} + \boldsymbol{A}\boldsymbol{b}_{\varepsilon'} + \boldsymbol{b} = \boldsymbol{x}' \simeq \boldsymbol{x}. \tag{19}$$

In other words, we only care about the specific region of the partition generated by the mixture of experts $a$RGB encoder that includes our pixel of interest. The flattened region $\tilde{\boldsymbol{x}}$ is decomposed into the center pixel $\boldsymbol{x}$ and the peripherals $B$.

$$h(\tilde{\boldsymbol{x}}) = h\left(\begin{bmatrix} \boldsymbol{x} \\ B \end{bmatrix}\right) = \boldsymbol{A}\boldsymbol{W}_{\varepsilon'}\begin{bmatrix} \boldsymbol{x} \\ B \end{bmatrix} + \boldsymbol{A}\boldsymbol{b}_{\varepsilon'} + \boldsymbol{b} \simeq \boldsymbol{x}. \tag{20}$$

This can be further decomposed if we decompose $\boldsymbol{W}_{\varepsilon'}$ into two matrices,

$$h(\tilde{\boldsymbol{x}}) = \boldsymbol{A}\begin{bmatrix} \boldsymbol{W}_{\mathrm{cen},\varepsilon'} & \boldsymbol{W}_{\mathrm{per},\varepsilon'} \end{bmatrix}\begin{bmatrix} \boldsymbol{x} \\ B \end{bmatrix} + \boldsymbol{A}\boldsymbol{b}_{\varepsilon'} + \boldsymbol{b}, \tag{21}$$

$$= \boldsymbol{A}\boldsymbol{W}_{\mathrm{cen},\varepsilon'}\boldsymbol{x} + \boldsymbol{A}\boldsymbol{W}_{\mathrm{per},\varepsilon'}B + \boldsymbol{A}\boldsymbol{b}_{\varepsilon'} + \boldsymbol{b} \simeq \boldsymbol{x}, \tag{22}$$

where $\boldsymbol{W}_{\mathrm{cen},\varepsilon'} \in \mathbb{R}^{C' \times C}$ is the effective weight for the center pixel $\boldsymbol{x}$, and $\boldsymbol{W}_{\mathrm{per},\varepsilon'} \in \mathbb{R}^{C' \times C(R^2-1)}$ is the effective weight for the peripheral pixels $B$.

Let us now assume a perfect lossless autoencoder $h^\star(\tilde{\boldsymbol{x}}) = \boldsymbol{x}$ for every combination of $\boldsymbol{x}$ and $B$. That is, the autoencoder $h^\star$ always returns the exact same pixel $\boldsymbol{x}$ without requiring $B$ to be a function of $\boldsymbol{x}$. In this case, equation 22 can be further decomposed into three equations:

$$\boldsymbol{A}\boldsymbol{W}_{\mathrm{cen},\varepsilon'}\boldsymbol{x} = \boldsymbol{x} \in \mathbb{R}^C, \tag{23}$$

$$\boldsymbol{A}\boldsymbol{W}_{\mathrm{per},\varepsilon'}B = \boldsymbol{0} \in \mathbb{R}^{C(R^2-1)}, \quad \text{and} \tag{24}$$

$$\boldsymbol{A}\boldsymbol{b}_{\varepsilon'} + \boldsymbol{b} = \boldsymbol{0} \in \mathbb{R}^C. \tag{25}$$

In particular, equation 23 should be satisfied for every combinations of RGB colors $\boldsymbol{x}$, and therefore we can conclude that $\boldsymbol{A}\boldsymbol{W}_{\mathrm{cen},\varepsilon'} = \boldsymbol{I}$ for the perfect autoencoder. Furthermore, equation 24 should be satisfied for every possible $B$, signifying that the encoder $f^\star$ of the perfect autoencoder $h^\star$ should project peripheral pixels to the nullspace of the decoder's weight $\boldsymbol{A}$, nullifying the information from the peripherals to propagate into estimating the pixel $\boldsymbol{x}$ at the center. In other words, the encoder $f$ does not learn to infer either $\boldsymbol{x}$ from $B$ or $B$ from $\boldsymbol{x}$ but simply acts as a separation of information between the center pixel and the peripheral pixels.

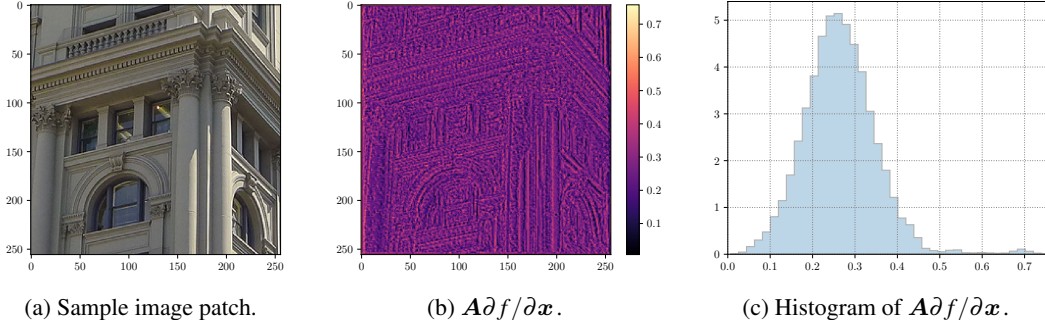

(a) Sample image patch.
(b) $A \partial f / \partial x$.
(c) Histogram of $A \partial f / \partial x$.

Figure 7: **Measurement of the degree of self-reference of the $a$RGB encoder.** The sample image is brought from Urban100 dataset (Huang et al., 2015). For a perfect autoencoder with no structure encoding capability, the values of $A \partial f / \partial x$ should be 1 for every pixel. However, in spite of our high reconstruction accuracy (62.738dB PSNR for this patch, which corresponds to an average color deviation of a pixel of 0.069/255), the value of $A \partial f / \partial x$ vary from 0.0066 to 0.7582, with average of 0.2654. This is unexpectedly low regarding the high accuracy, indicating that the reconstruction of our autoencoder heavily relies on the pixel's neighboring structure. Note that $A \in \mathbb{R}^{C \times C'}$ and $f / \partial x \in \mathbb{R}^{C' \times H \times W}$, where each element is obtained mutually independently. The heatmap and the histogram is obtained by taking the root-mean-square of the values over the three color channels.

In practice, however, this does not happen. In fact, the matrices $W_{\text{cen}, \varepsilon'}$ and $W_{\text{per}, \varepsilon'}$ are not just mathematical tools to represent the mechanism how the complicated nonlinear encoder $f$ acts upon the given input $x$. From equation 21, we can express only the encoder $f$ as:

$$f(\tilde{x}) = f\left( \begin{bmatrix} x \\ B \end{bmatrix} \right) = \begin{bmatrix} W_{\text{cen}, \varepsilon'} & W_{\text{per}, \varepsilon'} \end{bmatrix} \begin{bmatrix} x \\ B \end{bmatrix} + b_{\varepsilon'} = W_{\text{cen}, \varepsilon'} x + W_{\text{per}, \varepsilon'} B + b_{\varepsilon'}. \quad (26)$$

Taking the derivatives reveals interesting relationship between the gradients of $f$ and the effective weight at a particular pixel.

$$\frac{\partial f}{\partial x} = W_{\text{cen}, \varepsilon'} \quad \text{and} \quad \frac{\partial f}{\partial B} = W_{\text{per}, \varepsilon'}. \quad (27)$$

Therefore, for the lossless autoencoder, the relationship between the encoder's gradient and the decoder's weights are:

$$A \frac{\partial f}{\partial x} = I \quad \text{and} \quad A \frac{\partial f}{\partial B} B = 0. \quad (28)$$

We can calculate the matrix multiplication $A \partial f / \partial x$ and see how this differs from the identity to determine how the encoder mixes information between $x$ and $B$. Note that $\partial f / \partial x \in \mathbb{R}^{C' \times C}$ is a gradient only at the locus $c$. Figure 7 shows a simple experiment to check if the value of $A \partial f / \partial x$ deviates from unity. As the result presents, our $a$RGB encoder relies strongly on the neighboring structure to reconstruct a pixel.

# B IMPLEMENTATION DETAIL

This section presents additional details for training the main $a$RGB autoencoder used throughout the experiments in Section 3.

## B.1 TRAINING THE $a$RGB AUTOENCODER

**Architecture.** The $a$RGB autoencoder consists of three models: a convolutional router $f_r$, $K = 20$ convolutional experts $f_1, \ldots, f_K$, and a linear decoder $g$. The architecture is drawn in Table 5, 6, and 7, where C3 and C1 denote convolutions with kernel size 3 and 1, respectively, and L0.2 is a leaky ReLU with negative slope 0.2. Also, BN is a batch normalization with the same channel width as the output channel of the convolution in the same line. For $3 \times 3$ convolutions, we use zero padding of size 1. The router $f_r$ consists of three $3 \times 3$ and two $1 \times 1$ convolutions, with batch normalization (Ioffe and Szegedy, 2015) and leaky ReLU with negative slope 0.2 between each pair

Table 5: **Architecture of $f_{\mathbf{r}}$.**

| | |
|---|---|
| C3 → L0.2 | $3ch \to 64ch$ |
| C3 → BN → L0.2 | $64ch \to 128ch$ |
| C3 → BN → L0.2 | $128ch \to 256ch$ |
| C1 → BN → L0.2 | $256ch \to 512ch$ |
| C1 → Softmax | $512ch \to 20ch$ |

Table 6: **Architecture of $f_k$.**

| | |
|---|---|
| C3 → L0.2 | $3ch \to 32ch$ |
| C3 → L0.2 | $32ch \to 64ch$ |
| C3 → L0.2 | $64ch \to 128ch$ |
| C3 | $128ch \to 128ch$ |

Table 7: **Architecture of $g$.**

| | |
|---|---|
| C1 (no bias) | $128ch \to 64ch$ |
| C1 (no bias) | $64ch \to 32ch$ |
| C1 | $32ch \to 3ch$ |

of convolutions, except for after the first convolution, where we put only a leaky ReLU. Each expert $f_k$ for all $k \in \{1, \ldots, K\}$ has identical architecture, with four $3 \times 3$ convolutions and leaky ReLUs with negative slope of 0.2 in between. We do not use normalization layers in the experts. All the layers are initialized by the default PyTorch initializer (Paszke et al., 2019), meaning that each expert is initialized with different weights. Overall, the router $f_{\mathbf{r}}$ has a receptive field of size $7 \times 7$, and each expert $f_k$ has a receptive field of size $9 \times 9$, hence $R = 9$ in Section 2.2. For the linear decoder, we use three $1 \times 1$ convolutions without activations and normalization layers, with each layer gradually reducing the channel dimension. Only the last convolution of the decoder has a bias term. We have empirically found that this three linear layer architecture leads to a slightly better convergence. We can easily compute the effective weight of the decoder by multiplying all the internal weights with standard matrix multiplication. The decoder has receptive field of a single pixel and is also linear.

**Data preparation.** Our training dataset consists of a mix of image patches obtained from several sources, including DIV2K (Agustsson and Timofte, 2017), Flickr2K (Timofte et al., 2017), DIV8K (Gu et al., 2019), and ImageNet-1k (Russakovsky et al., 2015) datasets. The DIV2K, Flickr2K, and DIV8K datasets are high quality high resolution image datasets designed for training image super-resolution models. Those are selected in order to provide our autoencoder with rich structural information in clean, visually pleasing natural images. The DIV2K training set contains 800 high quality images, the Flickr2K set has 2,650 images, and the DIV8K dataset consists of 1,500 very high quality images up to 8K resolution. We also added the ImageNet-1k dataset to the training data to increase the diversity of image structures for supervision. Since the DIV2K, Flickr2K, and DIV8K datasets have much higher image size than one typically used for training a network, we have preprocessed these three dataset by cropping into patches of size $480 \times 480$ with stride 240. The results are 32,592 patches for the DIV2K, 107,403 patches for the Flickr2K, and 551,136 patches for the DIV8K dataset. The patch datasets are then concatenated with 1,281,167 images from the ImageNet-1k training data to build up the training data with 1,972,298 images in total. For each image fetched for training, a random crop of $256 \times 256$ is applied first, and then random horizontal and vertical flips, followed by a random 90 degrees rotation are applied consecutively.

**Finding the optimal hyperparameters.** We searched for the optimal architecture and other training hyperparameters by ablation studies mentioned in Section 4.4. For the ease of comparison, we unify the training process for the ablation studies in a simpler setting. Only using DIV2K dataset (Agustsson and Timofte, 2017) except for the second batch of experiments in Table 4, we vary the hyperparameters for training. Because the training dataset and the task become simple, we reduce the patch size to $192 \times 192$ and train our autoencoder for 300k iterations. Furthermore, since the per-pixel router $f_{\mathbf{r}}$ of our $a$RGB encoder distributes each pixel to a single expert, the number of iterations required for each expert to be trained using the same amount of data scales linearly to the number of experts. To compensate the effect, we changed the number of training iterations for the autoencoder accordingly. All the results in Section 4.4 is quantified by the accuracy of $\times 4$ image super-resolution task using RRDBNets (Wang et al., 2018b) trained to minimize the $L_1$ loss defined over the $a$RGB space. All the RRDBNet models are trained only with DIV2K dataset for 300k iterations from scratch.

**Training hyperparameters.** The weight of the load-balancing loss is selected to be $\lambda = 10^{-2}$ based on empirical observations. We found this to be the minimum value that ensures uniform expert assignment throughout the training process. This was chosen from a parameter sweep in the ranges from $10^0$ to $10^{-5}$ in powers of 10. The network is trained with a batch size of 16. An Adam (Kingma and Ba, 2015) optimizer is used with its default hyperparameters and an initial learning rate of 5e-4 is used. We use cosine learning rate schedule (Loshchilov and Hutter, 2017), starting from a period of 1k iteration, increased doubly up to 256k iterations. The training ends at 511k iterations.

Table 8: **Reconstruction accuracy of the $a$RGB autoencoder on out-of-distribution datasets.** We measured the PSNR scores *wihtout* quantization to 255 scale. Average RGB difference in the second row stands for the average absolute difference in the pixel's color value out of the maximum range of 255. All the values are significantly below the quantization gap of $0.5$, indicating almost perfect reconstruction of the input.

|  | Set5 | Set14 | Urban100 | DIV2K-Val | SIDD-Val | GoPro-Val |
|---|---|---|---|---|---|---|
| PSNR [dB] | 67.206 | 64.531 | 65.556 | 70.812 | 72.007 | 72.853 |
| Avg. RGB diff. | 0.0477 | 0.0602 | 0.0669 | 0.0418 | 0.0301 | 0.0266 |

**Autoencoding accuracy.** For completeness, we provide the autoencoding accuracy for the validation datasets used throughout this work in Table 8. Because the output of the $a$RGB autoencoder deviates from the original image only less than a single quantization step (1/255) on average, we regard the $a$RGB autoencoder as almost lossless in our analysis except for Appendix A, where we find that the nonideal reconstruction capability helps the autoencoder to learn structural prior of images.

### B.2 NOTES ON TRAINING IMAGE RESTORATION MODELS

**Change in optimal training hyperparameters** As emphasized throughout our manuscript, we have not changed the hyperparameters except for the very loss function in every experiment in Section 3. However, we also note that our demonstration does not necessarily mean that those setups are optimal. For instance, as mentioned in Section 4.3, altering the representation space leads to dramatic change in the scale and the shape of the gradients fed into the image restoration model during its training. Under stochastic gradient descent algorithms, this increase in the size of gradients of more than a hundredfold leads to significant changes in the training dynamics of those models. It is, therefore, less likely that the original set of hyperparameters is still optimal in the new representation space. Likewise, replacing the representation space may also change the optimal architecture for the image restoration task defined over the new space. Although we strongly believe that searching through the new possibilities allowed by our $a$RGB representation should be a fascinating research topic, this is beyond the scope of our paper. One example close to this direction is our demonstration of the NAFNet trained for $L_{1,aRGB}$ loss, reported in Section 3.3. In this experiment, because the replacement of metric causes the change in scale and shape of its gradients, we have doubled the weight of the metric for better convergence. As a result, this new setup has led us to better denoisers than both the original one and the one trained with only the representation space being altered. We leave further exploration of this topic for future work.

**Recommendation regarding gradient clipping** As a final remark, it is highly recommended to remove gradient clipping to maximize the advantage of using $a$RGB-based losses. Section 2.2 attributes the performance gain caused by the additive noise to the sensitivity increase in the $a$RGB encoder $f$. In practice, the effect can be observed as an increment of two orders of magnitude in the size of gradients of the image restoration models being trained. The same scale of optimizer's step size can only be achieved by increasing the learning rate a hundredfold, which quickly leads to training instability. We may safely conclude that the per-pixel distance losses in our $a$RGB space helps training of image restoration models by stably increasing the internal gradients. However, in recent image restoration techniques (Waqas Zamir et al., 2021; Chen et al., 2022), especially for the models with attention layers (Vaswani et al., 2017), gradient clipping is a common practice to stabilize the training of the model. To take advantage of our method, in Section 3, we changed gradient clipping mechanism to clamp at a inf norm of 20 for every experiment. This value is barely touched throughout the training process.

## C MORE QUANTITATIVE RESULTS ON PERCEPTUAL IMAGE SUPER-RESOLUTION

Table 9 extends Section 3.2 to provide full evaluation results on the $4\times$ perceptual super-resolution task using ESRGAN (Wang et al., 2018b). As mentioned in Section 3.2, our training process exactly follows the official implementation (Wang et al., 2018b). We first pre-train the network, RRDBNet, with DIV2K (Agustsson and Timofte, 2017) and Flickr2K (Timofte et al., 2017) combined, for 1M iterations using RGB $L_1$ loss. Then, the weights are fine-tuned with the loss written in the first column

Table 9: **Complete quantitative results for training ESRGAN $\times 4$ in the $a$RGB space.** Improved results are highlighted in **boldface** characters.

| Training objective | | Set14 | | | | | | B100 | | | | | |
|---|---|---|---|---|---|---|---|---|---|---|---|---|---|
| | | PSNR↑ | SSIM↑ | LPIPS↓ | DISTS↓ | NIQE↓ | FID↓ | PSNR↑ | SSIM↑ | LPIPS↓ | DISTS↓ | NIQE↓ | FID↓ |
| Pre-trained RRDBNet† | | 27.189 | 0.7551 | 0.2710 | 0.09212 | 6.1376 | 74.673 | 26.492 | 0.7216 | 0.3578 | 0.16287 | 6.3598 | 92.620 |
| $0.01L_1$ | $+0.005L_{\text{Adv}}$ | 24.629 | 0.6590 | 0.1823 | 0.07302 | 4.3218 | 62.582 | 24.084 | 0.6351 | 0.1805 | 0.10795 | 3.8023 | 57.491 |
| $0.01L_{1,a\text{RGB}}$ | $+0.005L_{\text{Adv}}$ | **24.955** | **0.6648** | **0.1431** | **0.06825** | **4.0553** | **52.020** | **24.423** | **0.6384** | **0.1683** | **0.10239** | **3.6126** | **52.355** |
| $0.01L_1$ | $+L_{\text{VGG}}+0.005L_{\text{Adv}}$† | 24.494 | 0.6543 | 0.1341 | 0.06374 | 3.8774 | 56.700 | 23.909 | 0.6205 | 0.1617 | 0.09603 | 3.6636 | 51.521 |
| $0.01L_{1,a\text{RGB}}$ | $+L_{\text{VGG}}+0.005L_{\text{Adv}}$ | **24.796** | **0.6623** | **0.1281** | **0.06052** | **3.7230** | **52.797** | **24.138** | **0.6306** | **0.1595** | **0.09578** | **3.4185** | **47.665** |

| Training objective | | Manga109 | | | | | | Urban100 | | | | | |
|---|---|---|---|---|---|---|---|---|---|---|---|---|---|
| | | PSNR↑ | SSIM↑ | LPIPS↓ | DISTS↓ | NIQE↓ | FID↓ | PSNR↑ | SSIM↑ | LPIPS↓ | DISTS↓ | NIQE↓ | FID↓ |
| Pre-trained RRDBNet† | | 29.654 | 0.8928 | 0.0972 | 0.00880 | 5.4084 | 12.190 | 25.496 | 0.7951 | 0.1963 | 0.02278 | 5.6236 | 23.729 |
| $0.01L_1$ | $+0.005L_{\text{Adv}}$ | 27.059 | 0.8349 | 0.0671 | 0.00759 | 3.4607 | 11.472 | 23.535 | 0.7373 | 0.1322 | 0.01911 | 3.9479 | 18.428 |
| $0.01L_{1,a\text{RGB}}$ | $+0.005L_{\text{Adv}}$ | 26.932 | 0.8255 | 0.0685 | 0.00806 | **3.4006** | **11.366** | 23.348 | 0.7204 | **0.1289** | 0.01990 | **3.8524** | **18.015** |
| $0.01L_1$ | $+L_{\text{VGG}}+0.005L_{\text{Adv}}$† | 26.441 | 0.8170 | 0.0646 | 0.01036 | 3.5758 | 11.282 | 22.776 | 0.7033 | 0.1232 | 0.02432 | 4.2067 | 20.616 |
| $0.01L_{1,a\text{RGB}}$ | $+L_{\text{VGG}}+0.005L_{\text{Adv}}$ | **26.651** | **0.8186** | **0.0630** | **0.00863** | **3.4245** | **10.907** | **23.270** | **0.7196** | **0.1183** | 0.02050 | **3.8982** | **17.739** |

| Training objective | | DIV2K-Val | | | | | | OST300 | | | | | |
|---|---|---|---|---|---|---|---|---|---|---|---|---|---|
| | | PSNR↑ | SSIM↑ | LPIPS↓ | DISTS↓ | NIQE↓ | FID↓ | PSNR↑ | SSIM↑ | LPIPS↓ | DISTS↓ | NIQE↓ | FID↓ |
| Pre-trained RRDBNet† | | 29.466 | 0.8306 | 0.2537 | 0.00190 | 5.4860 | 15.910 | 25.960 | 0.7218 | 0.3565 | 0.10544 | 6.2643 | 41.673 |
| $0.01L_1$ | $+0.005L_{\text{Adv}}$ | 27.102 | 0.7687 | 0.1282 | 0.00248 | 3.0419 | 13.593 | 23.613 | 0.6311 | 0.1856 | 0.06141 | 3.5593 | 22.749 |
| $0.01L_{1,a\text{RGB}}$ | $+0.005L_{\text{Adv}}$ | **27.218** | 0.7622 | **0.1235** | **0.00240** | 3.0896 | **12.936** | 23.587 | 0.6212 | 0.1989 | 0.06762 | **3.4612** | 24.603 |
| $0.01L_1$ | $+L_{\text{VGG}}+0.005L_{\text{Adv}}$† | 26.627 | 0.7033 | 0.1154 | 0.00545 | 3.0913 | 13.557 | 23.249 | 0.6166 | 0.1688 | 0.05734 | 3.3834 | 21.851 |
| $0.01L_{1,a\text{RGB}}$ | $+L_{\text{VGG}}+0.005L_{\text{Adv}}$ | **26.845** | **0.7500** | **0.1110** | **0.00351** | **2.9615** | **12.799** | **23.649** | **0.6338** | **0.1662** | **0.05484** | 3.5247 | **19.952** |

† The official ESRGAN models (Wang et al., 2018b).

for 400k iterations using DIV2K, Flickr2K, and OSTv2 (Wang et al., 2018a) datasets combined. Preparation of the datasets are done with the same code the authors has provided. In this task, the $a$RGB loss is a simple $L_1$ loss over the range of the $a$RGB encoder. We leave all the other training hyperparameters untouched.

In addition to LPIPS (Zhang et al., 2018a), NIQE (Mittal et al., 2013), and FID (Heusel et al., 2017) metrics, we added DISTS (Ding et al., 2020), a pairwise perceptual metric for image restoration. Moreover, four common benchmarks for the percpeptual super-resolution, *i.e.*, Set14 (Zeyde et al., 2010), BSD100 (Martin et al., 2001), Manga109 (Matsui et al., 2017), and OutdoorSceneTest300 (Wang et al., 2018a) datasets are added for comparison. The updated results are shown in Table 9. The reported values are calculated in the RGB domain, cropping 4 pixels from the outer sides for PSNR, SSIM, and NIQE (Mittal et al., 2013) scores. For DISTS (Ding et al., 2020) and LPIPS (Zhang et al., 2018a), we use the code from official repo, and for FID (Heusel et al., 2017), we report scores from the PyTorch reimplementation (Seitzer, 2020) of the original Tensorflow code.

The data shows that $L_{1,a\text{RGB}}$ loss without the VGG perceptual loss yields comparable results to the $L_1$ loss without the perceptual loss in adversarial training. However, when equipped with VGG perceptual loss, the results show a general increase in performance over the original ESRGAN in all the distoration-based metrics (PSNR, SSIM), the pairwise perceptual metrics (LPIPS, DISTS), and the unpaired quality assessment metrics (NIQE, FID).

# D  MORE QUALITATIVE RESULTS

This section provides more visual results from the main experiments. Figure 8 through 13 show results from the ESRGAN models (Wang et al., 2018b) trained with and without $a$RGB representation. Empirically, the supervision given by our $a$RGB encoder helps the model avoid generating visual artifacts and color inconsistency induced by adversarial training. Figure 14 shows additional results from the real image denoising task solved with NAFNet (Chen et al., 2022). Lastly, image deblurring using MPRNet (Waqas Zamir et al., 2021) trained from our $a$RGB representation is further demontrated in Figure 15 and 16. As visualized in Appendix E, the additional information embodied in the extra dimensions of the $a$RGB representation resembles edgeness information of an image. We conjecture that this allows a pairwise per-pixel distance defined over our $a$RGB space to provide image restoration models with stronger supervision leading to the reconstruction of sharper edges. The results reveal that this effect is realized as suppression of artifacts in the perceptual image super-resolution task, sharper produced images in the image denoising task, and better reconstruction of edges and more accurate alignments in the image deblurring task.

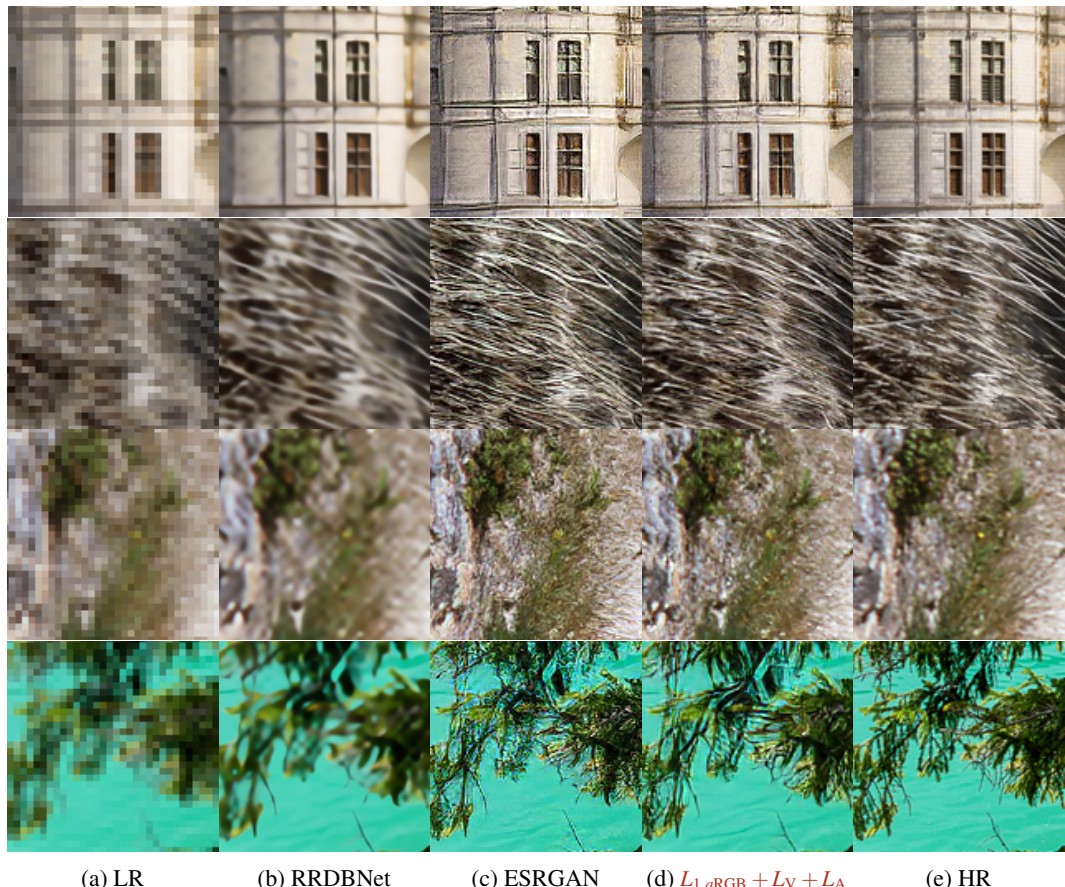

| (a) LR | (b) RRDBNet | (c) ESRGAN | (d) $L_{1,a\text{RGB}} + L_\text{V} + L_\text{A}$ | (e) HR |

Figure 8: **Qualitative comparison of ESRGAN models trained with different loss functions on DIV2K-Val (Agustsson and Timofte, 2017) benchmark.**

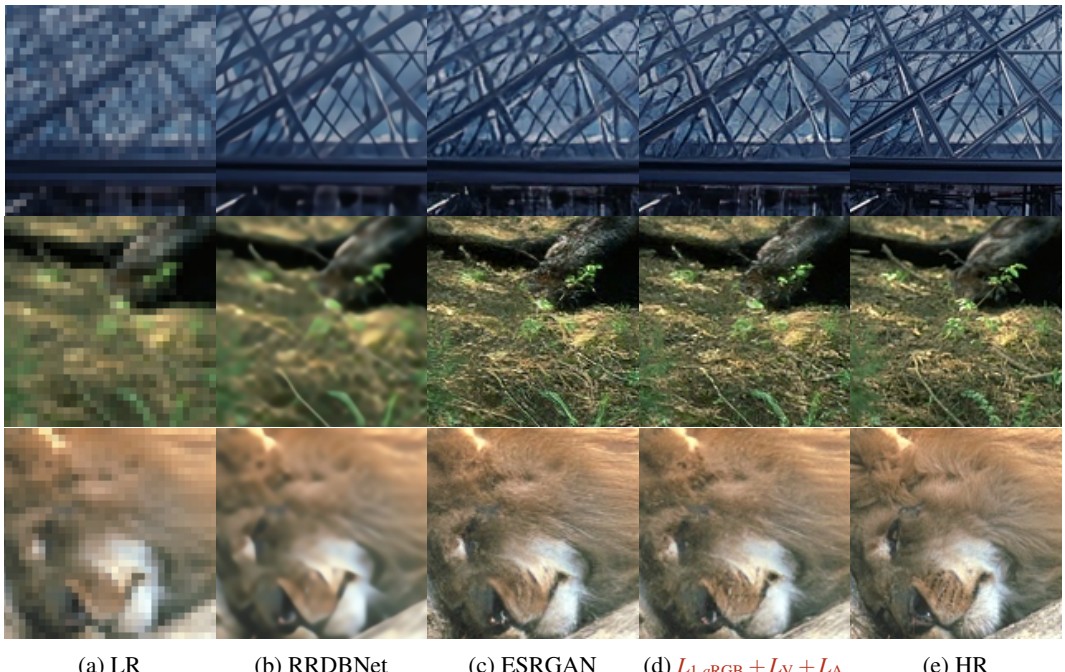

| (a) LR | (b) RRDBNet | (c) ESRGAN | (d) $L_{1,a\text{RGB}} + L_\text{V} + L_\text{A}$ | (e) HR |

Figure 9: **Qualitative comparison of ESRGAN models trained with different loss functions on B100 (Martin et al., 2001) benchmark.**

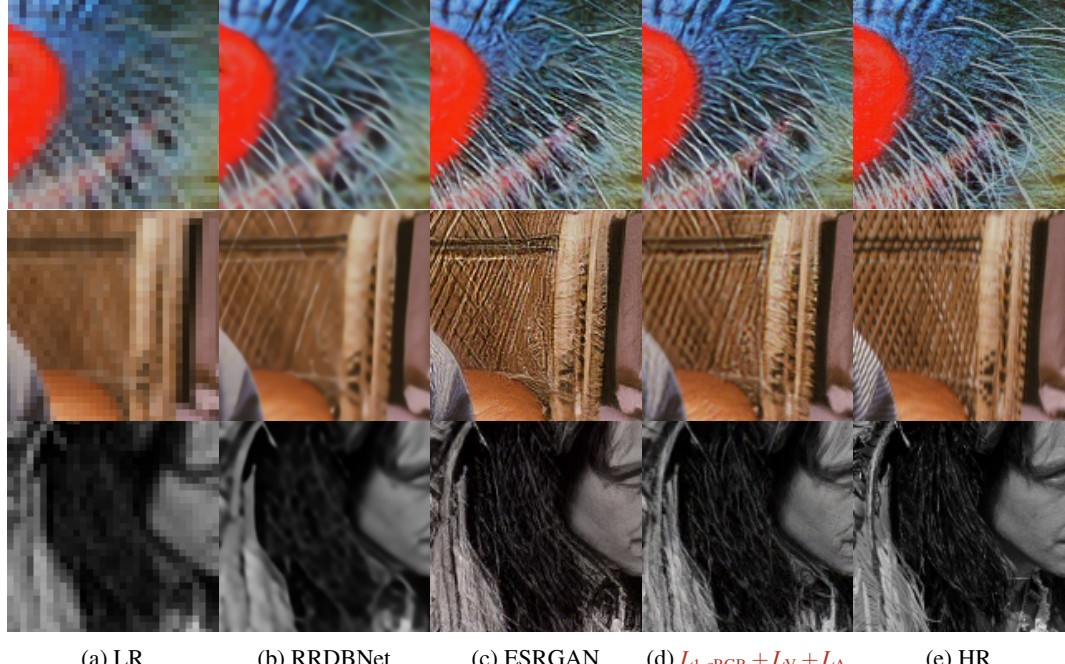

| (a) LR | (b) RRDBNet | (c) ESRGAN | (d) $L_{1,a\mathrm{RGB}} + L_\mathrm{V} + L_\mathrm{A}$ | (e) HR |

Figure 10: **Qualitative comparison of ESRGAN models trained with different loss functions on Set14** (Zeyde et al., 2010) **benchmark.**

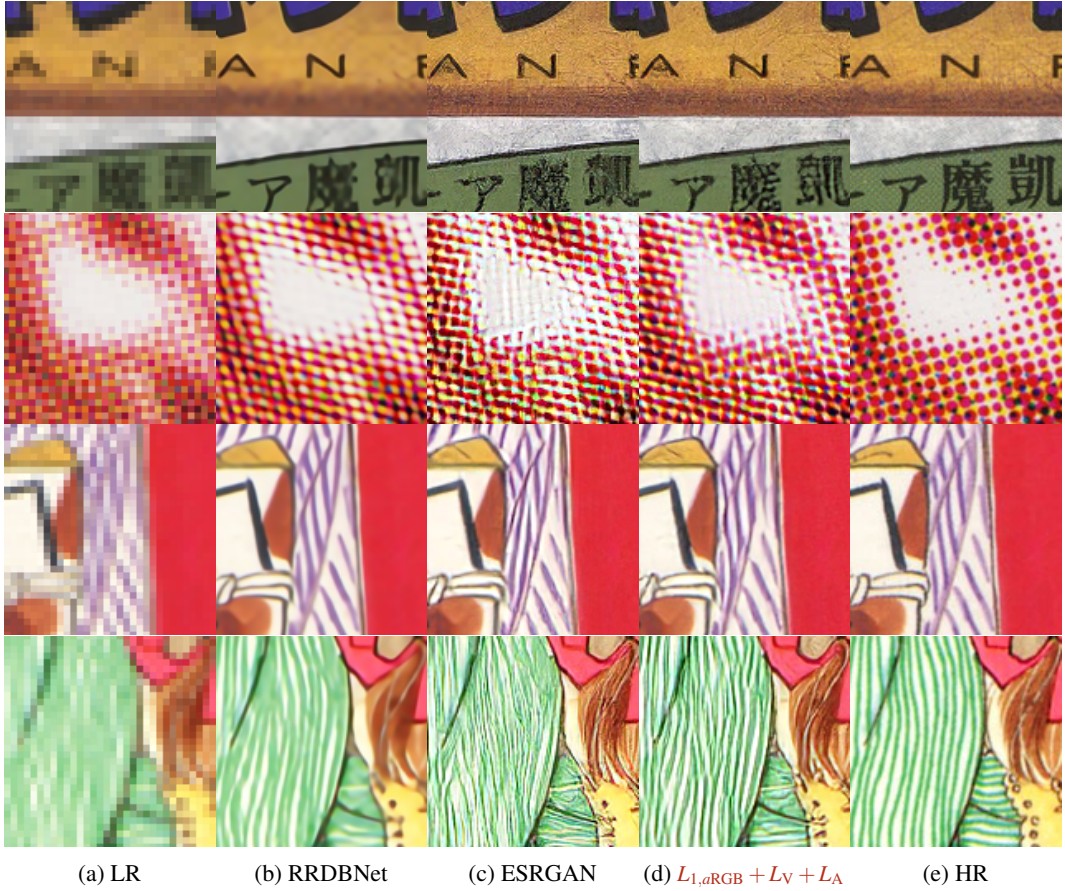

| (a) LR | (b) RRDBNet | (c) ESRGAN | (d) $L_{1,a\mathrm{RGB}} + L_\mathrm{V} + L_\mathrm{A}$ | (e) HR |

Figure 11: **Qualitative comparison of ESRGAN models trained with different loss functions on Manga109** (Matsui et al., 2017) **benchmark.**

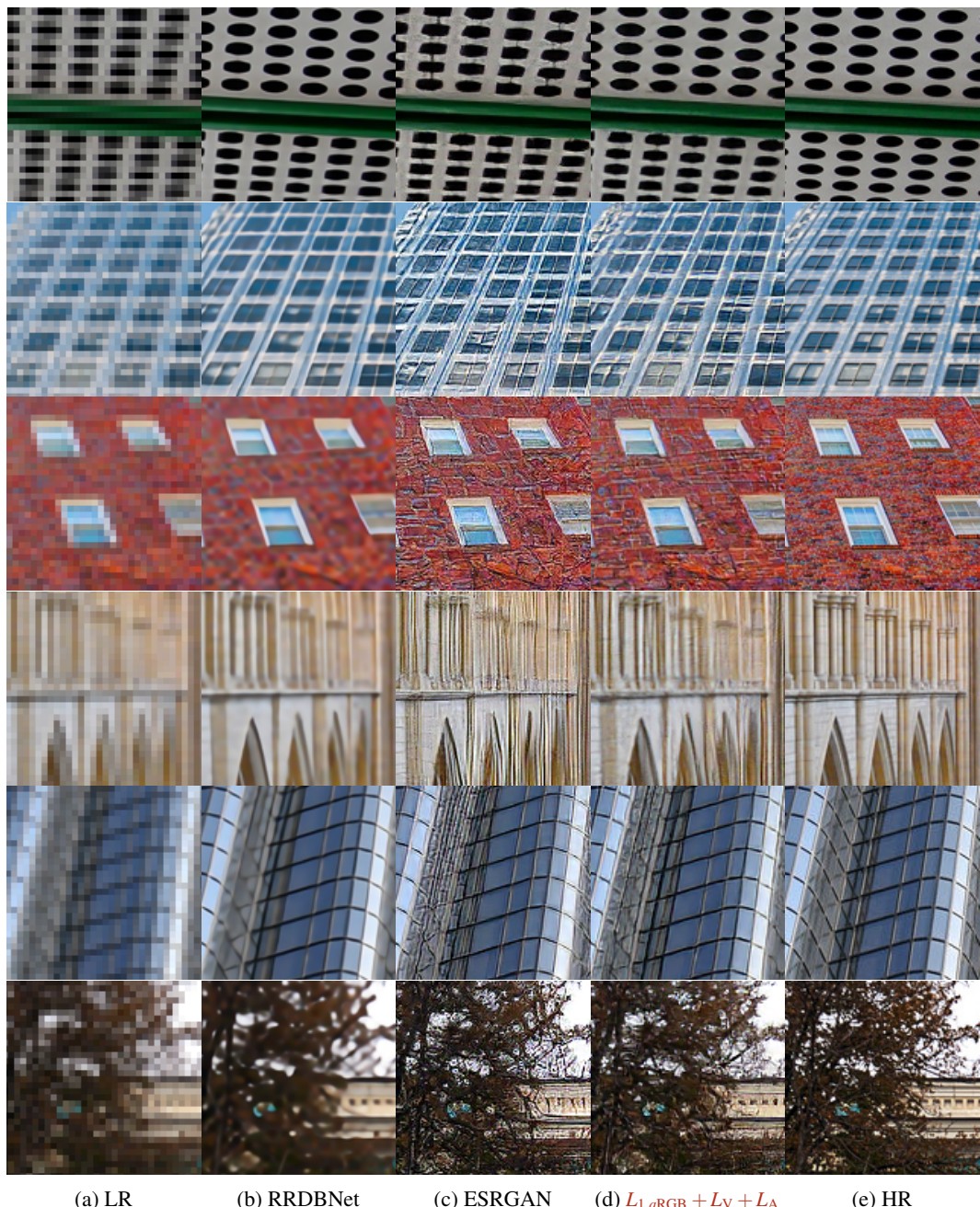

(a) LR         (b) RRDBNet         (c) ESRGAN         (d) $L_{1,a\text{RGB}} + L_V + L_A$         (e) HR

Figure 12: **Qualitative comparison of ESRGAN models trained with different loss functions on Urban100 (Huang et al., 2015) benchmark.**

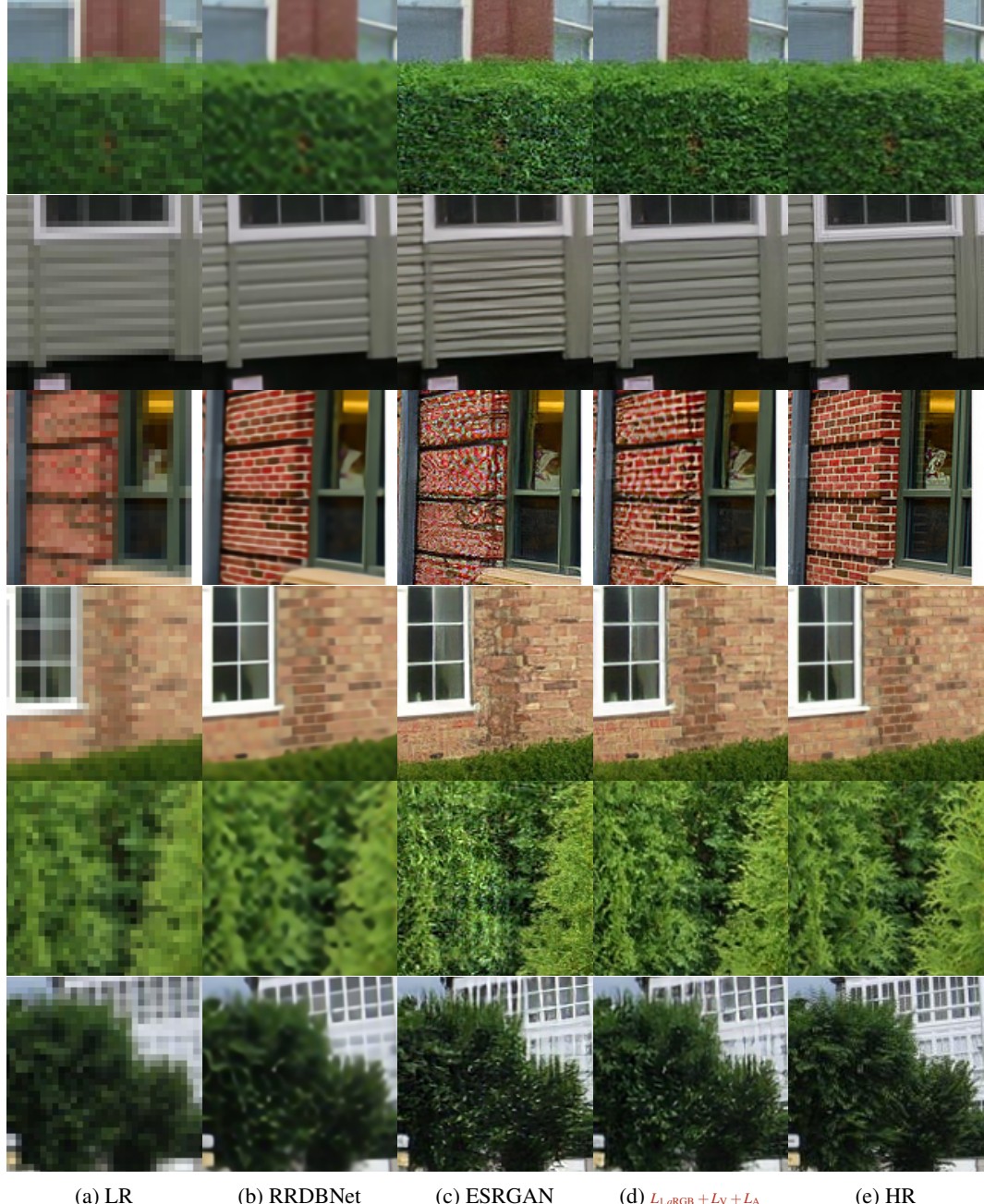

(a) LR    (b) RRDBNet    (c) ESRGAN    (d) $L_{1,aRGB} + L_V + L_A$    (e) HR

Figure 13: **Qualitative comparison of ESRGAN models trained with different loss functions on Out-doorSceneTest300 (Wang et al., 2018a) benchmark.**

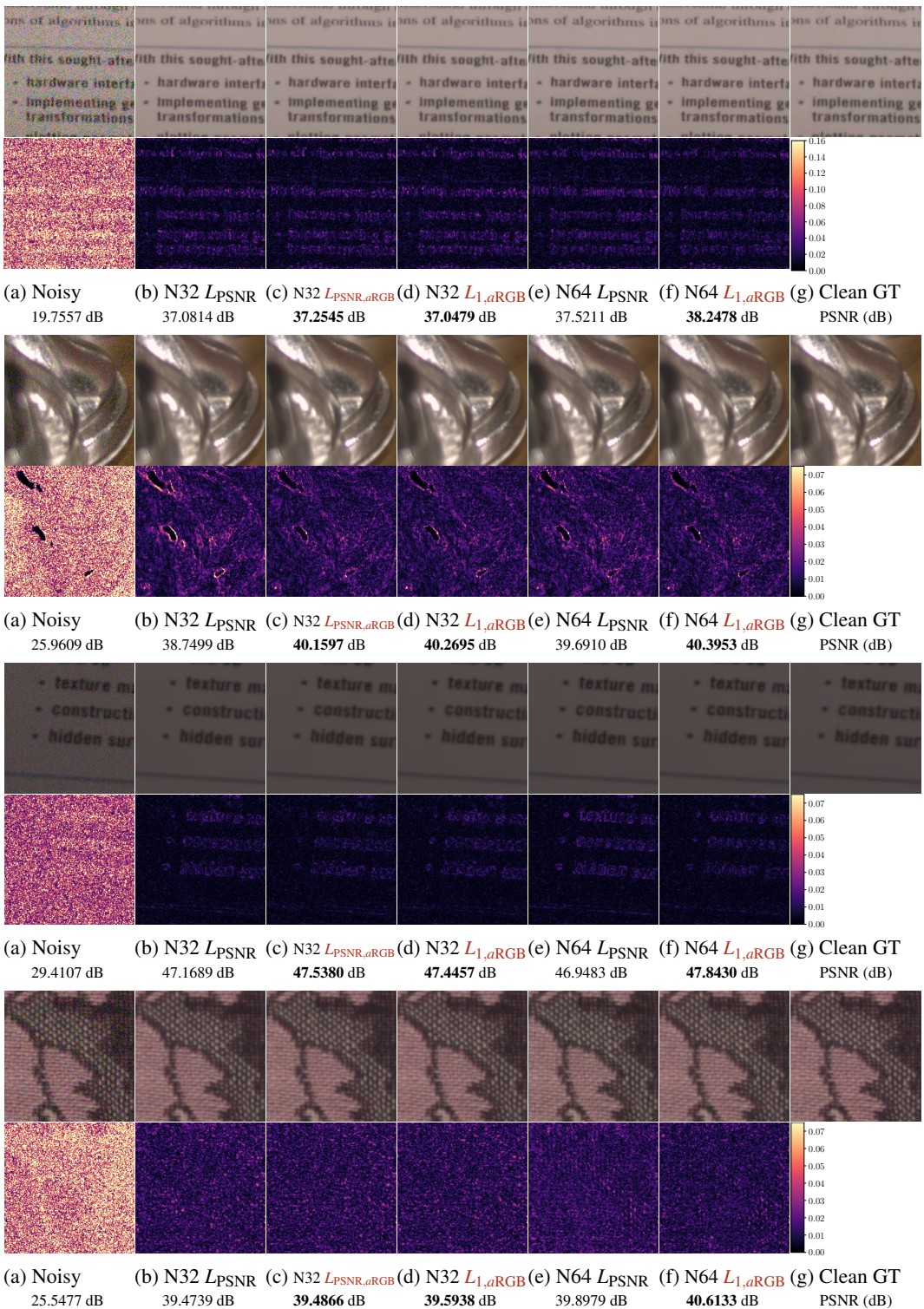

Figure 14: **Qualitative comparison of real image denoising models on SIDD benchmark (Abdelhamed et al., 2018).** Each column corresponds to each row in Table 2. N32 corresponds to NAFNet-width32 and N64 corresponds to NAFNet-width64. The bottom rows show the maximum absolute difference in color with a range of $[0, 1]$.

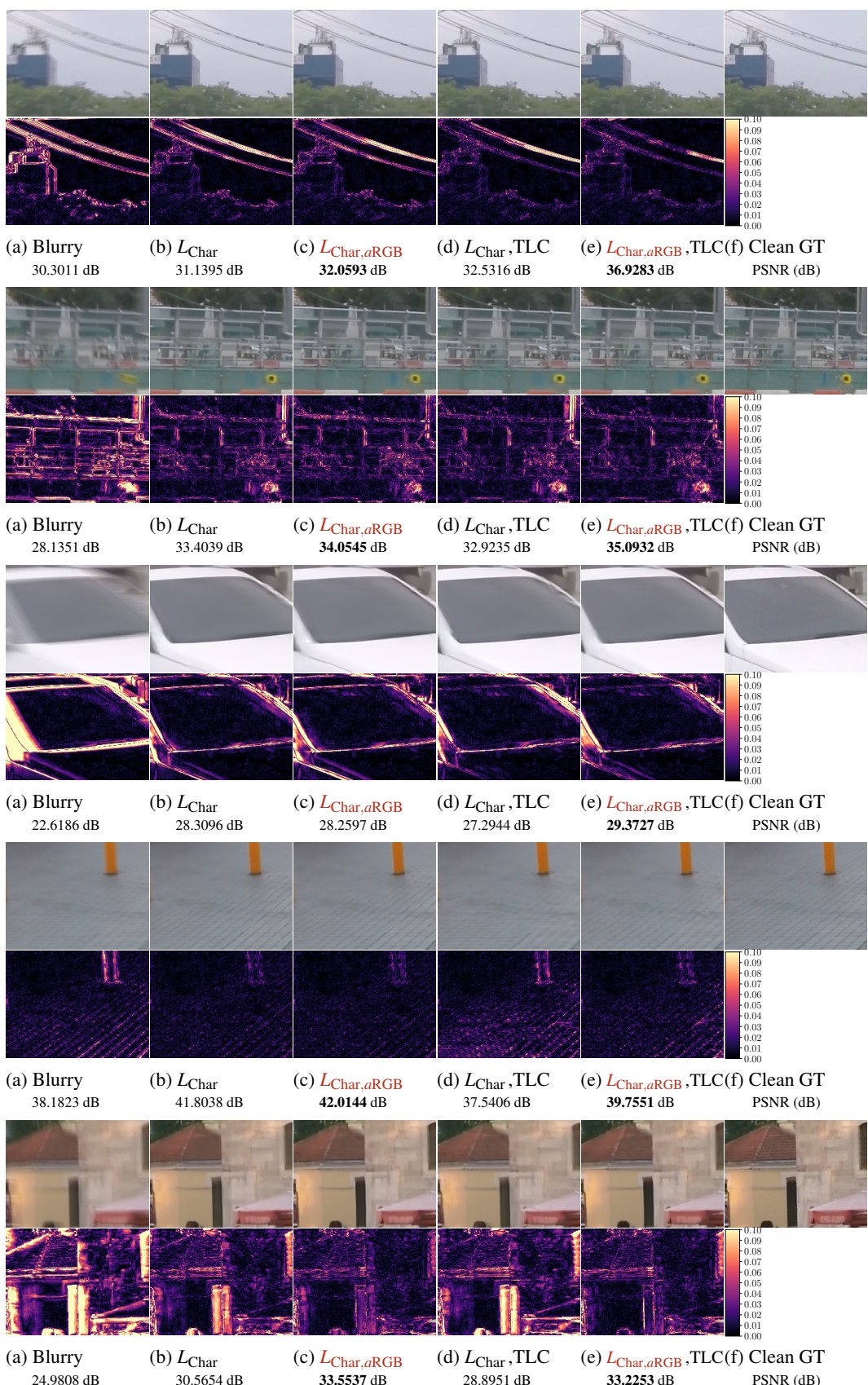

Figure 15: **Qualitative comparison of motion blur deblurring models in GoPro benchmark (Nah et al., 2017).** Each column corresponds to each row in Table 3. The bottom rows show the maximum absolute difference in color with a range of $[0, 1]$.

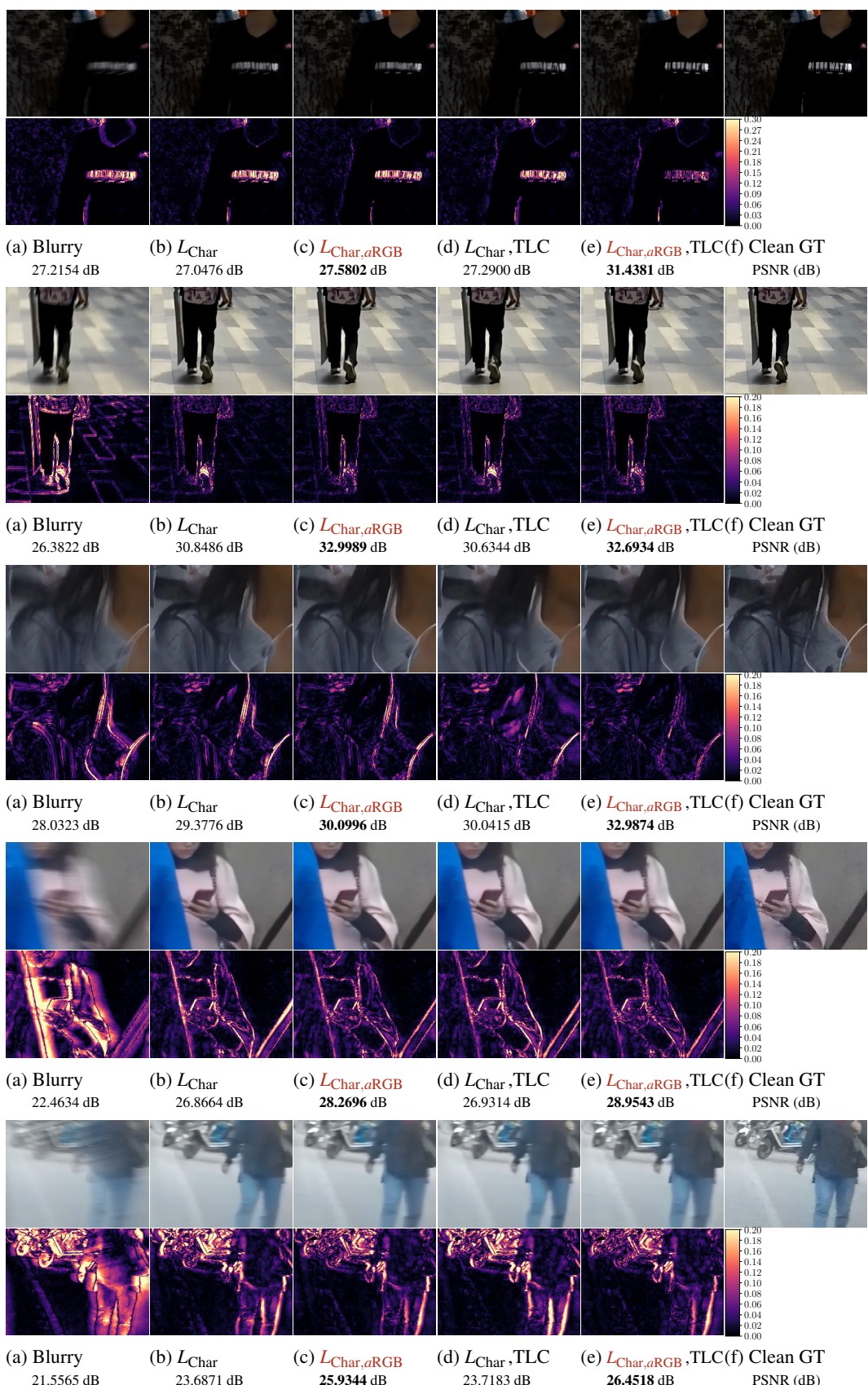

Figure 16: **Qualitative comparison of motion blur deblurring models in HIDE benchmark (Shen et al., 2019).** Each column corresponds to each row in Table 3. The bottom rows show the maximum absolute difference in color with a range of $[0, 1]$.

# E  UNDERSTANDING THE *a*RGB REPRESENTATION SPACE

This section is divided into three parts: In the first part of this section, we discuss further on the embedding decomposition test conducted in Section 4.1. Mixing different types of images in the same way as we did in Section 4.1 reveals how our *a*RGB representation encodes additional information in the extra dimensions. Next, we show more examples of the t-SNE (van der Maaten and Hinton, 2008) visualization of our learned embeddings in addition to Figure 5c. Unlike segmentation maps typically generated and assessed for semantic tasks, the ones produced by our *a*RGB encoder are extremely fine-grained and complicated, yet we find some common structural features in the results. In the last part, we conduct another experiment to visualize how each expert in the *a*RGB encoder *f* learns to specialize.

## E.1  DECOMPOSITION OF *a*RGB REPRESENTATION

In Section 4.1, we have discussed how the linearity of our decoder *g* helps understanding the learned representation space. In particular, given its linear weight $A$ and bias $b$, we can decompose any given embedding in the *a*RGB representation $\xi \in \mathbb{R}^C$ into a sum of two orthogonal vectors:

$$\xi = A^\dagger A \xi + (I - A^\dagger A)\xi, \tag{29}$$

where we denote the matrix $A^\dagger$ as the Moore-Penrose pseudoinverse of $A$. We can regard the multiplicand of the second term $I - A^\dagger A$ as a linear projection operator onto the nullspace of $A$, and the multiplicand of the first term $A^\dagger A$ as a linear projection onto the orthogonal complement of the nullspace of $A$. Mathematically, and also empirically, it is easy to see that summation of any vector projected onto the nullspace of $A$ leads to no change in the decoded image $x' = g(\xi)$. That is,

$$g(\xi + (I - A^\dagger A)\zeta) = A(\xi + (I - A^\dagger A)\zeta) + b \tag{30}$$

$$= A\xi + b + A(I - A^\dagger A)\zeta \tag{31}$$

$$= A\xi + b + (A - AA^\dagger A)\zeta \tag{32}$$

$$= A\xi + b + (A - A)\zeta \tag{33}$$

$$= A\xi + b \tag{34}$$

$$= g(\xi). \tag{35}$$

The identity in the fourth row is from the equality $A = AA^\dagger A$ of Moore-Penrose pseudoinverse. Therefore, the subspace to which the projection operator $I - A^\dagger A$ is mapped is the allowed degree of freedom that additional information can be embedded.

Inversion of mixed embedding uncovers what information lies in this particular subspace. In addition to the results in Figure 5a, Figure 17 and 18 show visual results of *a*RGB embedding inversion. The flat image source is manually synthesized by a single color gradient. The real image source is a patch brought from DIV2K validation dataset (Agustsson and Timofte, 2017), and the Gaussian noise source is sampled from $\mathcal{N}(0.5, 0.5)$, where the RGB range is $[0, 1]$. Each patch has an equal size of $256 \times 256$. First, we obtain the two *a*RGB embeddings from both sources. Then, we take the parallel part $f_\parallel$ from the first source and the perpendicular part $f_\perp$ from the second source. An image is optimized to produce the synthesized embedding as its *a*RGB representation. Due to the simple training setup, all the optimization quickly converges after 50 iterations of SGD with learning rate 0.1, as shown in Figure 19. We show the result of the optimization in Figure 17. The edge structure is extracted and highlighted by applying the Laplacian operator on the final image and displayed in Figure 18.

From the results, we can conclude the followings: Firstly, the parallel part of the *a*RGB embedding dominates the color information. This is predictable since this parallel component is the information that is decoded back to the RGB image with our linear decoder *g*. Secondly, the perpendicular part of the *a*RGB embedding conveys edge structure from the source image. As a consequence, we can retrieve this information from the *a*RGB representation following the same process used to train the inversion in this section. Moreover, this edgeness is the additional information brought with our *a*RGB representation to serve as a performance boost for training the image restoration models. As our main results in Section 3 suggests, the reduction of visual artifacts in perceptual image super-resolution and the enhancement of edge structures in image denoising and deblurring

are attributed to this particular information carried in the perpendicular embeddings. Lastly, we can clearly observe that this additional information is diminished away if the underlying image is highly noisy, far from the manifold of clean, natural images. From this observation, we can conclude that the information learned by the $a$RGB autoencoder from its training dataset gives the model a structural prior in order to process images similar to its original training data. This argument aligns with our finding in the ablation studies in Section 4.4, where a network trained from different dataset generally produce poorer restoration models.

## E.2 TOPOLOGY OF THE LEARNED $a$RGB SPACE

We have conjectured in Section 2.1 that the underlying distribution of small image patches are disconnected, exhibiting very complex structures. We can indirectly check the validity of our argument by visualizing the learned embeddings of an image using dimension reduction techniques. Specifically, we use t-SNE (van der Maaten and Hinton, 2008) algorithm to embed $H \times W = 65,536$ vectors of size $C = 128$ into a two dimensional plane to visualize the structure of our learned embedding for a particular image patch.

In this small experiment, we expect two outcomes: First, we expect perceptually distinct groups of embedding vectors to be appeared in the t-SNE results. Although the geometric information of the 2D projection carries little meaning in depicting the exact structure of the underlying feature space, visually distinguishable clustering in this region will signify that the structure of underlying representation is not connected into a single manifold, but consists of multiple disconnected regions with distinct characteristic values. Second, we expect that the experts are well specialized, requiring that clusters of $a$RGB *pixels* each assigned to a single expert to be appeared in the embeddings. Figure 20 shows three more examples in addition to Figure 5b and 5c from our main manuscript. The sample images are picked from different datasets, *i.e.*, DIV2K (Agustsson and Timofte, 2017), Urban100 (Huang et al., 2015), and Manga109 (Matsui et al., 2017), having different color distributions, contents, and styles. However, as the results show that the embeddings generated from this set of images have several commonalities: First, the embeddings are clustered in a well-separated regions. We can observe two distinct types of clusters: *common* groups where multiple experts are involved in generating similar embeddings and *expert-specific* groups where a single expert dominates in encoding the information of pixels. Existence of the first type of groups indicates the existence of common subspace between the feature spaces of each specialized expert. The latter type of groups show a clear evidence of expert specialization in our $a$RGB encoder. From the observations, we conclude that our initial design philosophy of the network serves its original purpose.

## E.3 VISUALIZATION OF THE LEARNED FEATURES OF THE $a$RGB ENCODER

In this last section of our paper, we provide another visualization results to facilitate understanding of the behavior of our $a$RGB encoder. In order to visualized the learned features for individual expert of our encoder, we simply maximize the activation of a single last channel of one of the experts. Starting from a random image of size $32 \times 32$, which is more than three times larger than the receptive field of our network, we run a simple maximization of the average activation of each channels to produce a single feature image for each channel of an expert. Figure E.3 shows some of the results. First, we observe that channels of the same index at each of the experts are maximally activated at the similar color distribution. This similarity comes from the shared linear decoder of our autoencoder. However, we also notice that the same set of filters are maximally stimulated at different *patterns*. The results uncover another evidence of expert specialization in our $a$RGB autoencoder.

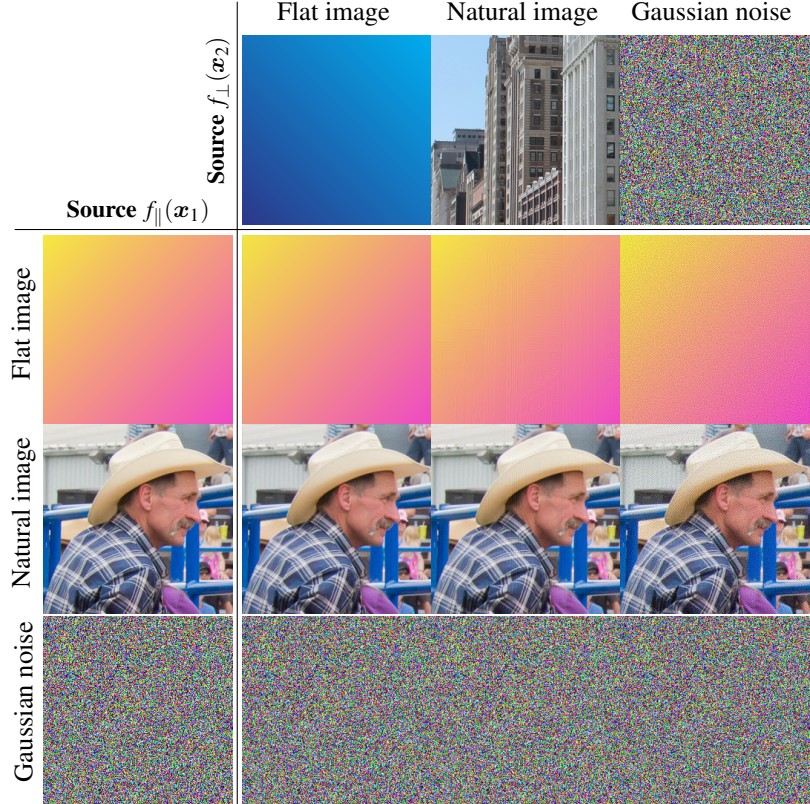

Figure 17: **Decomposition of the *a*RGB representation space.** The *a*RGB embeddings of the two groups of images are decomposed into orthogonal components and mixed together $\boldsymbol{\xi}_{\text{mix}} = f_{\parallel}(\boldsymbol{x}_1) + f_{\perp}(\boldsymbol{x}_2)$. The matrix shows images that best matches the synthesized *a*RGB embedding $f^{-1}(\boldsymbol{\xi}_{\text{mix}})$.

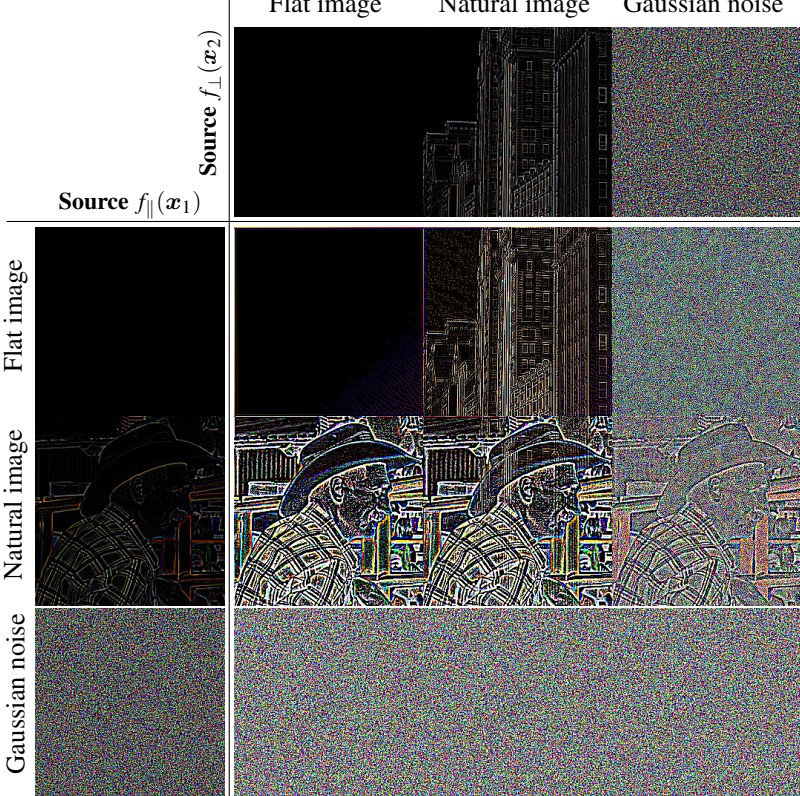

Figure 18: **Edge-enhanced inversion results of Figure 17.** A discrete Laplacian operator is applied to the same images in Figure 17 to enhance the high-frequency structures for clearer understanding. The results reveal that the perpendicular component of the *a*RGB embedding $f_{\perp}$ contributes to high-frequency structures.

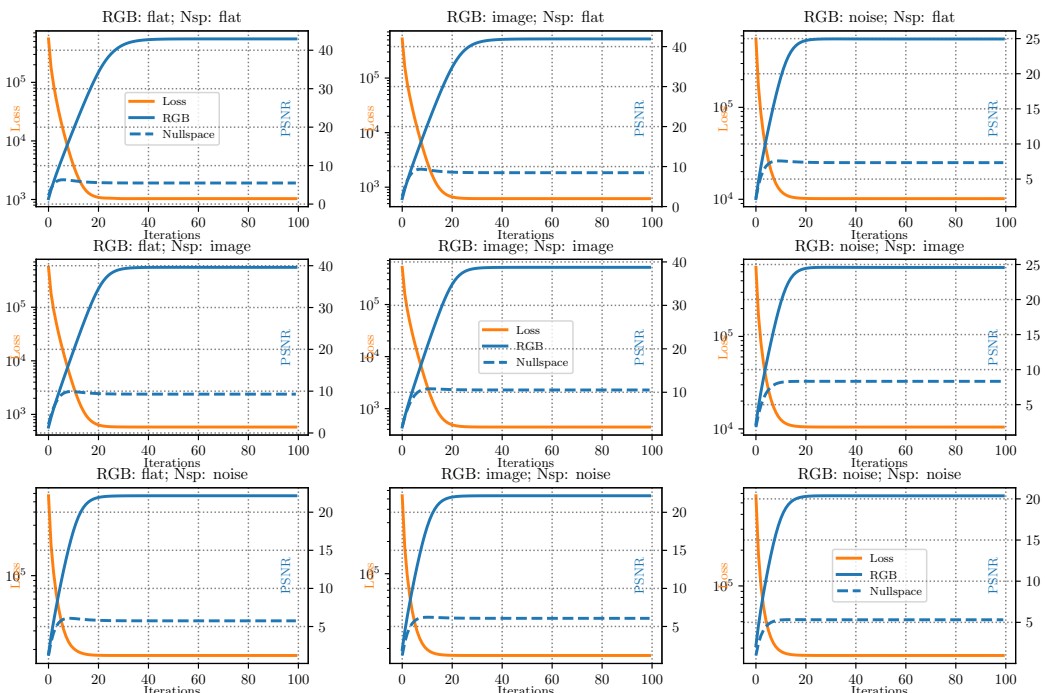

Figure 19: **Training curve for the decomposition test.** All the embedding inversion test quickly converge after 50 iterations. *RGB* corresponds to the source image $\boldsymbol{x}_1$ used for the parallel component $\boldsymbol{\xi}_{\parallel} = \boldsymbol{A}^{\dagger}\boldsymbol{A}f(\boldsymbol{x}_1) = \boldsymbol{A}^{\dagger}\boldsymbol{A}\boldsymbol{\xi}_{\text{mix}}$ of the target $a$RGB embedding $\boldsymbol{\xi}_{\text{mix}}$, and *Nullspace* corresponds to the source image $\boldsymbol{x}_w$ used for the perpendicular component $\boldsymbol{\xi}_{\perp} = (\boldsymbol{I} - \boldsymbol{A}^{\dagger}\boldsymbol{A})f(\boldsymbol{x}_2) = (\boldsymbol{I} - \boldsymbol{A}^{\dagger}\boldsymbol{A})\boldsymbol{\xi}_{\text{mix}}$, where $m\boldsymbol{A}$ is the weight of the linear decoder $g$. As shown in Figure 17, The low-frequency color distribution of the resulting inversion follows that of the parallel component's source $\boldsymbol{x}_1$, resulting in high PSNR scores. Although the PSNR scores between the inversions $f^{-1}(\boldsymbol{\xi}_{\text{mix}})$ and the corresponding source images $\boldsymbol{x}_2$ of the perpendicular component $f_{\perp}(\boldsymbol{x}_2)$ are low, Figure 18 reveals that the perpendicular components encode high frequency information of the image.

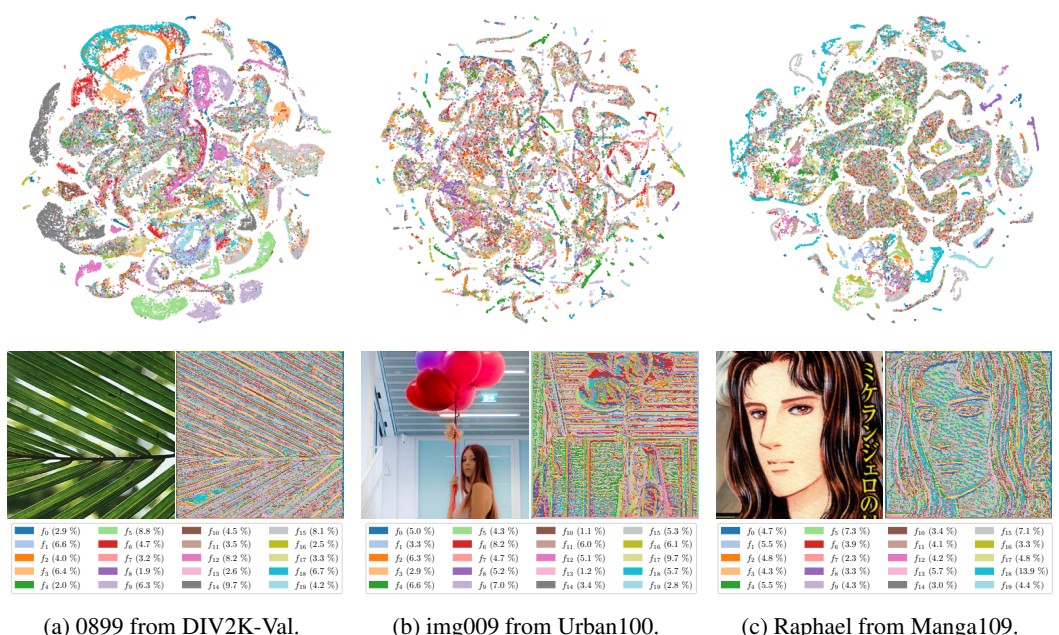

(a) 0899 from DIV2K-Val.  (b) img009 from Urban100.  (c) Raphael from Manga109.

Figure 20: **More examples on expert specialization using t-SNE and segmentation map.** Sample images are brought from three well-used super-resolution benchmark datasets, *i.e.*, DIV2K (Agustsson and Timofte, 2017), Urban100 (Huang et al., 2015), and Manga109 (Matsui et al., 2017). Although the content and the style of each patches are widely different, the distribution of the learned *a*RGB embeddings in these patches exhibit similar pattern: the distributions are decomposed into *common* groups, where multiple experts are involved in the encoding, and *expert-specific* groups.

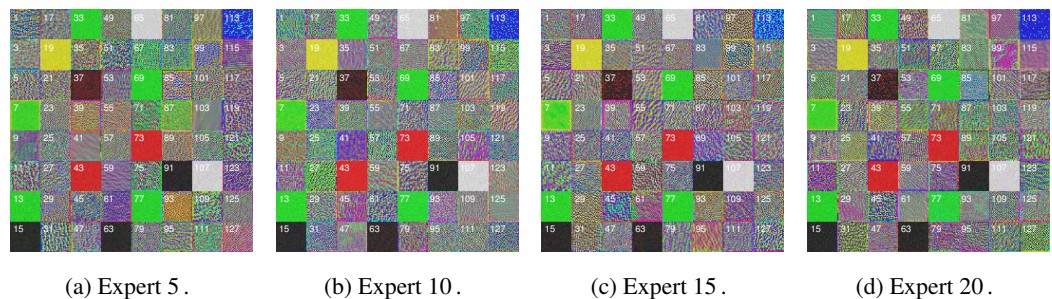

(a) Expert 5 .  (b) Expert 10 .  (c) Expert 15 .  (d) Expert 20 .

Figure 21: **Visualization of the output filters of the experts.** Randomly initialized $32 \times 32$ images are trained to maximize a specific filter at the last convolutional layer of the selected expert. The ID of each filter is annotated with white numbers. Note that the *a*RGB representation space has a dimension of 128 , the same as the number of filters in the last layer of each experts. The results show that while filters of different experts encoding the same channel is maximally activated at a similar average color, the high-frequency patterns each filter maximally attends to vary significantly.

