# OpenReview forum: "Rethinking RGB Color Representation for Image Restoration Models"
_ICLR.cc/2024/Conference — Submitted to ICLR 2024_

### Official Review · Reviewer_cKyU · 2023-10-27

**Soundness:** 2 fair
**Presentation:** 3 good
**Contribution:** 3 good
**Rating:** 5
**Confidence:** 4

**Summary:**

The paper addresses limitations in the RGB color representation when conveying local image structures. The authors introduce an augmented RGB (aRGB) space, developed using an encoder, which captures both color and structural details. This new space offers more freedom in selecting loss functions and showcases performance improvements in various image processing tasks. Additionally, the aRGB space enhances interpretability, with the authors providing a comprehensive analysis of its properties and benefits.

**Strengths:**

This paper proposed an augmented RGB (aRGB) space is the latent space of an autoencoder that comprises a single affine decoder and a nonlinear encoder, trained to preserve color information while capturing low-level image structures. The results imply that the RGB color is not the optimal representation for image restoration tasks.

**Weaknesses:**

Based on the experiments, compared to previous methods, the improvement brought by vggloss is quite limited, with an increase of 0.1dB (PSNR) in Table 1 and 0.02dB in Table 2. Moreover, it hasn't been compared with other perceptual methods, such as lpips or ssim loss.

Although this paper claims to introduce a method that doesn't calculate loss in the RGB domain, the loss function used in training still falls within the category of pixel-based feature scale. Overall, it represents a relatively minor improvement to the loss function for low-level vision. Hence, the performance enhancement is limited.

Is the selection of the number of "experts" highly dependent on experience? Will different tasks have significant variations? It seems that an inappropriate selection of the number of experts might lead to even lower performance than not using this loss function at all.

**Questions:**

See weakness.

---

> ### Author Response · Authors · 2023-11-17
>
> We thank you for sparing your valuable time to review our work and raise important and intriguing questions. To provide comprehensive answers, we would like to categorize your concerns in four questions. Please feel free to notify us if we have misinterpreted your comments. Due of the length constraint, our comment is divided in two.
>
> ### **[Q1: Significance of the performance improvement]**
>
> > Based on the experiments, compared to previous methods, the improvement brought by vggloss is quite limited, with an increase of 0.1dB (PSNR) in Table 1 and 0.02dB in Table 2.
>
> | Loss | GoPro PSNR | GoPro SSIM |
> | --- | --- | --- |
> | L1-RGB | 30.40 | 0.9018 |
> | L1-$a$RGB | 30.85 (+0.45 dB) | 0.9096 (+0.0078) |
>
> - First, as a response to your concern in the performance gain of our $a$RGB representation space, we have conducted a new experiment on another deblurring method [4*]. The results in the table above show consistent improvement.
>
> - While performance gains are sometimes marginal (e.g., 0.02 dB in Table 2), we would like to note that 0.1 dB difference in PSNR in Table 1 is significant, since PSNR is measured in log10 scale. Furthermore, including the table above, we have shown consistent performance improvement across various restoration tasks and models, with the same pre-trained $a$RGB autoencoder. We believe the results indicate that our $a$RGB space can serve as a promising alternative to the RGB space.
>
> - We would like to highlight that our main focus in this work is to test whether a pixel-wise color difference is a necessary loss function in various image restoration tasks. Please note that the results in both tables are obtained from the models that do not use the RGB representation space for calculating the loss functions, and yet show comparable (often better) performance in terms of RGB PSNR. We believe these results raise research questions as to whether RGB color space should be must-use space for restoration tasks, hopefully introducing an interesting research direction.
>
> ### **[Q2: Dependence of the task in the optimal number of experts]**
>
> > Is the selection of the number of "experts" highly dependent on experience? Will different tasks have significant variations? It seems that an inappropriate selection of the number of experts might lead to even lower performance than not using this loss function at all.
>
> - In addition to our ablation study in Table 4 and Section 4.4 in the manuscript, we provide an additional ablation study with a deblurring model [4*]. The results are shown in the table below. Results demonstrate that **the number of experts in our system is indeed critical for the final performance.** However, the **tendency of such influence is similar among different tasks**, showing that the advantage of $a$RGB is maximized when using 20 experts.
>
> | Loss | GoPro PSNR | GoPro SSIM |
> | --- | --- | --- |
> | L1-RGB (Reference) | 30.40 | 0.9018 |
> | L1-$a$RGB (20 experts) | 30.85 **(+0.45 dB)** | 0.9096 **(+0.0078)** |
> | L1-$a$RGB (5 experts) | 29.14 **(-1.26 dB)** | 0.8133 **(-0.0885)** |

---

> ### Author Response · Authors · 2023-11-17
>
> (Continued from the last comment)
>
> ###  **[Q3: Concerning the difference between the RGB and the $a$RGB spaces]**
>
> > Although this paper claims to introduce a method that doesn't calculate loss in the RGB domain, the loss function used in training still falls within the category of pixel-based feature scale.
>
> - To our understanding, your concern in this part is that **how can we call our representation space not the RGB even if it does have the same spatial scale in its features**. We believe your question highlights the key difference between the scale and the content of image features.
>
> - As you have pointed out, we have designed our representation space to have the same spatial dimensions with the image. However, we have increased the channel dimensions from 3 (RGB) to 128. Overall, the feature dimension of our $a$RGB encoder is 128 X H X W, larger than that of the input image. Our aim was to introduce **more information** in our representation space **for each individual pixel**, rather than to remove unnecessary information from it. Therefore, we have dubbed our representation space **augmented** RGB. Our representation space **includes** the information of the RGB colors as its (linear) subspace. We believe that our $a$RGB space-based losses are different from the RGB space-based ones, not because our $a$RGB space represents completely different information from the colors, but because it **extends** the pixel-wise color information.
>
> - **We have revised the manuscript in Section 2.1 accordingly** to better clarify our claim.
>
> ### **[Q4: On the performance of perceptual loss]**
>
> > Moreover, it hasn't been compared with other perceptual methods, such as lpips or ssim loss.
>
> - Please understand if we have misread your intention in this part. To our understanding, you have interpreted our $a$RGB encoder’s role similar to the VGG-Net in the monumental perceptual loss. However, we wish to point out that their roles are greatly different. The key is that our $a$RGB representation space **replaces the RGB space** for loss calculation, whereas perceptual losses, such as VGG loss, LPIPS loss, DISTS loss, differentiable SSIM loss, and the dual-space losses such as the Fourier space loss [2] all require RGB distances to guide the training. In other words, **the $a$RGB space-based distance losses and perceptual losses do not compete with each other**. Rather, they are **complementary** in their roles.
>
> - Table 1 and Section 3.2 in our manuscript show one possible collaboration of these two types of losses in perceptual super-resolution task. Here, existence of the VGG loss helps our $a$RGB loss stabilize the training of ESRGAN [3].
>
> ### **[Concluding remark]**
>
> Finally, we thank you for your recognition of our main claim and contributions. Although we have tried our best providing answers to each of your concerns, we might have missed a point or two. Please understand if we have done so and feel free to raise any issue. We are always open to new discussions.
>
> ### **[Reference]**
> - [4*] Seungjun Nah et al., “Deep Multi-Scale Convolutional Neural Network for Dynamic Scene Deblurring,” In *CVPR* 2017.

---

### Official Review · Reviewer_tXWj · 2023-10-30

**Soundness:** 2 fair
**Presentation:** 3 good
**Contribution:** 2 fair
**Rating:** 3
**Confidence:** 5

**Summary:**

The authors propose aRGB loss for image restoration. The proposed loss is defined based on the latent space of an autoencoder, which consists of a single affine decoder and a nonlinear encoder. The autoencoder is trained to preserve color information while capturing low-level image structures. The authors replace per-pixel losses in the RGB space with their counterparts in training various image restoration models such as deblurring, denoising, and perceptual super-resolution.

**Strengths:**

+ A new latent representation space is proposed and employed for restoration loss design.
+ The aRGB loss is defined for diverse image restoration tasks.

**Weaknesses:**

-In the paper, the performance of the proposed loss are demonstrated on perceptual SR task. The results in table 1 are confusing. The PSNR and SSIM of RRDBNet are the highest among all the settings, but they are not bolded. The SSIM of the last setting is worse than most of settings for DIV2K-Val dataset, but it is bolded as better score.
-For perceptual SR and image deblur tasks, there are considerable baselines perform better than ESRGAN and MPRNet. For example, restormer and NAFNet could be used for deblurring evaluation. In this way, we can test whether the proposed loss could consistently boost performance and lead to a new SOTA.
-The performance gains are too small, which can hardly verify the effectiveness of the proposed loss.

**Questions:**

The proposed loss is similar to Fourier loss, which is also decomposed the image upon pre-defined basis. Can authors discuss the difference between them and compare their performance?

---

> ### Author Response · Authors · 2023-11-17
>
> We express our deepest gratitude for your constructive review and the time and effort you have invested for reviewing our work. We are excited to discuss about your considerate concerns. Due to the length constraint of this comment section, we split our response into two.
>
> ### **[W1: Regarding Table 1]**
>
> > In the paper, the performance of the proposed loss are demonstrated on perceptual SR task. The results in table 1 are confusing. The PSNR and SSIM of RRDBNet are the highest among all the settings, but they are not bolded. The SSIM of the last setting is worse than most of settings for DIV2K-Val dataset, but it is bolded as better score.
>
> - We appreciate your constructive feedback. We have changed Table 1 accordingly to clarify the main points we wanted to make with it.
>
> - In Table 1 and Section 3.2 of the main manuscript, our goal was to excel in perceptual metrics in perceptual image super-resolution. In particular, we would like to note that we focus on the improvement brought by replacing the L1 loss in the RGB space with the L1 loss in our $a$RGB space.
>
> - The first row stands for the PSNR-oriented reference model just for the reference. This is not trained under the perceptual super-resolution setting. The rows two to five represent methods to enhance perceptual metrics. Here, we intended to make comparisons between the second and third rows, while comparing the fourth and fifth rows.
>
> - We hope this change removes the original confusion.
>
> ### **[W2-3: Significance of the denoising and deblurring results]**
>
> > For perceptual SR and image deblur tasks, there are considerable baselines perform better than ESRGAN and MPRNet. For example, restormer and NAFNet could be used for deblurring evaluation. In this way, we can test whether the proposed loss could consistently boost performance and lead to a new SOTA.
> > The performance gains are too small, which can hardly verify the effectiveness of the proposed loss.
>
> - Regarding state-of-the-art methods, we agree with your concerns that the goal of designing a loss function is to work with the best models we have at that time. However, please understand that due to the strict resource constraint upon us, we had to find the optimal combination of experiments to best demonstrate the versatility of our method in various tasks and model types.
>
> - Regarding our experiments for perceptual super-resolution in Section 3.2 of our manuscript, due to heavy computation of Restormer and NAFNet, many previous works (ESRGAN [5*], Real-ESRGAN [8*], SPSR [9*], LDL [10*], CAL-GAN [11*]) adopt the RRDB network as their baseline. Moreover, we would like to note that NAFNet-width64, the state-of-the-art model you have mentioned is used for the denoising task, yet not for the deblurring task, in Section 3.3 of our manuscript.
>
> - From the experiments in Section 3 of the manuscript, we can observe consistent performance gain in various tasks and model architectures. The improvements were achieved using the same pre-trained $a$RGB autoencoder. Therefore, we would like to argue that our $a$RGB space suggests that there can be alternatives to the RGB space for the guidance of training image restoration models.
>
> - As a final remark, though it is not a state-of-the-art, we hope we can justify our approach more clearly with an additional experiment. Here, as an answer to your regard on our small performance gain, we apply our method on another deblurring model [4*], as presented in the table below. Consistent with Table 2 in the main manuscript, our method demonstrates superior performance. The numbers are calculated from the author’s official code repository.
>
>
> | Loss | GoPro PSNR | GoPro SSIM |
> | --- | --- | --- |
> | L1 | 30.40 | 0.9018 |
> | L1-$a$RGB | 30.85 (+0.45 dB) | 0.9096 (+0.0078) |

---

> ### Author Response · Authors · 2023-11-17
>
> (Continued from the last comment)
>
> ###  **[Q1: Difference from Fourier loss]**
>
> > The proposed loss is similar to Fourier loss, which is also decomposed the image upon pre-defined basis. Can authors discuss the difference between them and compare their performance?
>
> - We find three key differences between the Fourier space loss and the per-pixel loss in our $a$RGB space.
>
>     1. Fourier space loss is typically combined with its primary loss, i.e., the RGB space-based loss, without removing the necessity of the RGB loss. This is the common practice of the Fourier loss we can found in other notable works [6*, 7*], too. However, in our work, **we suggest an alternative to the RGB space** with our $a$RGB space.
>
>     2. Fourier space loss calculates distances in the discrete dual of the image space by transforming the pair of images using FFT algorithm. FFT is known to be a linear operation, and therefore a Fourier space loss can be thought of as a linearly weighted distances between two image patches of fixed size. On the other hand, **our $a$RGB encoder is nonlinear**, and therefore the distance metrics are contracted and expanded based on the local patch structure. This unique feature of ours helps our $a$RGB-based loss functions adaptively weigh important structures. The evidence of the metric warping of our $a$RGB space is shown in Figure 5d.
>
>     3. Although the Fourier space loss can handle long range nonlocal correlations via resonance in Fourier domain, their resonance frequency is strictly fixed because the linear weight of the FFT algorithm is a constant. In contrast, our $a$RGB encoder **only cares for the local structure** within its receptive fields, yet it can **encode more complex structures adaptively** thanks to its Mixture-of-Experts architecture. Figures 5b, 5c and 20 in our manuscript show that pixel assignments to each of $a$RGB encoder’s expert depends on the local structures of its neighborhood.
>
> - To demonstrate the effectiveness of our $a$RGB loss over the Fourier space loss, we design an experiment to train a deblurring model [4*] using two types of losses: (1) the combination of the primary (the L1-RGB) and the dual (the L1-Fourier space loss) losses following recent works [6*, 7*] and (2) the L1-$a$RGB loss without using the L1-RGB loss. Please understand that the time and resources permitted to us is limited, so we are temporarily reporting the intermediate scores. However, the difference between the two method clearly demonstrates the superirority of our $a$RGB loss.
>
> | DeepDeblur / GoPro (100 epochs) | PSNR (dB) | SSIM |
> | --- | --- | --- |
> | Fourier space loss + L1-RGB | 27.0007 | 0.8241 |
> | L1-$a$RGB | 27.7348 (+0.73 dB) | 0.8416 (+0.0175) |
>
> ###  **[Concluding remark]**
>
> We hope this answer fits you well. We would like to conclude our remark with our deepest thanks to your acknowledgement to our novelty in designing the representation space and the versatility of our demonstration.
>
>
> ### **[Reference]**
> - [4*] Seungjun Nah et al., “Deep Multi-Scale Convolutional Neural Network for Dynamic Scene Deblurring,” In *CVPR* 2017.
> - [5*] Xintao Wang et al., “ESRGAN: Enhanced Super-Resolution Generative Adversarial Networks,” In *ECCVW* 2018.
> - [6*] Sung-Jin Cho et al., “Rethinking Coarse-to-Fine Approach in Single Image Deblurring,” In *ICCV* 2021.
> - [7*] Yuning Cui et al., “Selective Frequency Network for Image Restoration,” In *ICLR* 2023.
> - [8*] Xintao Wang et al., “Real-ESRGAN: Training Real-World Blind Super-Resolution with Pure Synthetic Data,” In *ICCVW* 2021.
> - [9*] Cheng Ma et al., “Structure-Preserving Image Super-Resolution,” In *T-PAMI* 2021.
> - [10*] Jie Liang et al., “Details or Artifacts: A Locally Discriminative Learning Approach to Realistic Image Super-Resolution,” In *CVPR* 2022.
> - [11*] JoonKyu Park et al., “Content-Aware Local GAN for Photo-Realistic Super-Resolution,” In *ICCV* 2023.

---

### Official Review · Reviewer_nF5y · 2023-11-03

**Soundness:** 3 good
**Presentation:** 4 excellent
**Contribution:** 3 good
**Rating:** 8
**Confidence:** 5

**Summary:**

The paper introduces a novel approach to address the limitations of per-pixel RGB distances in image restoration. The authors propose a new representation space called augmented RGB (aRGB) space, where each pixel captures neighboring structures while preserving its original color value. By replacing the RGB representation with aRGB space in the calculation of per-pixel distances, the authors demonstrate performance improvements in perceptual super-resolution, image denoising, and deblurring tasks. In addition, the aRGB space allows for better interpretability through comprehensive analysis and visualization techniques. The contributions of this paper lie in the introduction of a versatile representation space, performance improvements in various image restoration tasks, and interpretability.

**Strengths:**

- The paper introduces the augmented RGB (aRGB) space for better image restoration.
- The paper provides a comprehensive and insightful analysis and visualization techniques for the aRGB space, enhancing interpretability. The analysis is solid and convincing.
- The versatility of the aRGB space allows for more freedom in choosing the loss function.

**Weaknesses:**

- The performance improvement of the proposed aRGB space in the denoising and debluring tasks seems insignificant. In Table 2, comparing the first two rows, and the last two rows, the PSNR gains are only 0.02 dB and 0.03 dB, respectively. In Table 3, the PSNR improvements between the last two rows are 0.07 dB on GoPro and 0.02 dB on HIDE dataset.

Additional comments
- Equation 6, L_{pair} should be L_{pixel}

**Questions:**

- The space aRGB is originally designed to encode structure information on a pixel basis. Why it can exhibit suppression of artifacts in SR tasks?
- The training process of aRGB auto-encoder does not involve any loss regarding local structure. Is it possible that the encoder also learns other information, e.g., texture, style?

---

> ### Author Response · Authors · 2023-11-17
>
> We deeply appreciate your positive review with comprehensive acknowledgement on our key contributions. We will try our best clearing your remaining questions on our work. Since our comment exceeds the character limit, we are splitting it into two.
>
> ### **[Comment 1: Typo]**
>
> > Equation 6, L_{pair} should be L_{pixel}
>
> We thank for your correction. We have revised the equation in the updated pdf.
>
> ### **[Q1: Explanation of the suppression of artifacts originated from adversarial training]**
>
> > The space aRGB is originally designed to encode structure information on a pixel basis. Why it can exhibit suppression of artifacts in SR tasks?
>
> - We believe that there are at least two reasons the artifacts are well suppressed by our $a$RGB encoder.
>
>     1. **Local information propagation**: Through the $a$RGB encoder, each pixel receives gradients from the pixel difference of its neighboring pixels. Visual artifacts can be seen as a disruption in high-frequency variation among adjacent pixels [12*]. The information propagation through our $a$RGB-based loss suppresses not only each pixel’s regression error, but each local region’s structural variations. This is not like in the conventional RGB space-based loss, which only cares about each pixel’s regression error. This analysis is further justified by our discussion in Appendix A. Here, we have shown that the amount of information propagation, i.e., the scale of gradients from adjacent pixels, is significant in our $a$RGB encoder.
>
>     2. **Edge encoding**: We can also consider this phenomenon from the representation perspective. As visually demonstrated in Figures 5a, 17, and 18, the $a$RGB encoder appends local edge information to each pixel. This means that the $a$RGB embedding space effectively has channels responsible for edge information. This structural part is directly compared by a per-pixel $a$RGB loss, suppressing undesired **structure** presented in the restoration output.
>
> - In short, we believe that the cause of artifact suppression you have mentioned is due to $a$RGB encoder’s **stronger guidance to minimize errors in high-frequency local structures**.
>
> ### **[Q2: Demonstration of the information encoded in the $a$RGB autoencoder]**
>
> > The training process of aRGB auto-encoder does not involve any loss regarding local structure. Is it possible that the encoder also learns other information, e.g., texture, style?
>
> - As you may have noticed, our $a$RGB autoencoder embeds simple local neighborhood structures such as edges at each pixel as visualized in Figures 5a, 17, and 18. We have also tried to clarify the sources of this structural information by decomposing the $a$RGB space in Section 4.1.
>
> - To our understanding, your concern is beyond this simple neighboring structures. It depends on how people define textures and styles. Nevertheless, we doubt that our $a$RGB space encodes **semantic** level of textures, for example, a checkerboard or long hairy textures. We have two reasons to believe so.
>
>     1. **Small receptive fields**: Our $a$RGB encoder has a receptive field size of 9x9. This size is sufficient to suppress high-frequency artifacts generated by other auxiliary losses as demonstrated in Table 1, and to enhance the edge structures in deblurring and denoising as suggested in results in Appendix D. However, this is not sufficient for the encoder to perceive semantic textures. Thus, we believe that our $a$RGB encoder encodes local structure by appending local edge information to each pixel within neighborhood of small area corresponding to its receptive field.
>
>     2. **Empirical reasons**: Although we have not included in our manuscript, we have conducted a simple experiment on training a single 1x1 convolutional layer to return VGG features from the $a$RGB embedding. If this experiment succeeds, we can conclude that the textual information up to VGG features’ level of abstract is indeed embedded in our $a$RGB space. However, this quickly has failed when we try to mimic the features from after a second downscaling block of the VGGNet.
>
> - To sum up, **the advantage of our $a$RGB space is not based on the semantic embedding**. In other words, the $a$RGB space-based loss functions and the ones derived from deeper networks, e.g., perceptual losses and adversarial losses, are **complements rather than substitutes**. We believe that experimental results in Table 1 of our manuscript supports this claim.

---

> ### Author Response · Authors · 2023-11-17
>
> (Continued from the last comment)
>
> ### **[W1: Significance of the denoising and deblurring results]**
>
> > The performance improvement of the proposed aRGB space in the denoising and debluring tasks seems insignificant. In Table 2, comparing the first two rows, and the last two rows, the PSNR gains are only 0.02 dB and 0.03 dB, respectively. In Table 3, the PSNR improvements between the last two rows are 0.07 dB on GoPro and 0.02 dB on HIDE dataset.
>
> - We acknowledge your concerns. We have conducted additional experiment on another deblurring model [4*], changing its original L1-RGB loss with our L1-$a$RGB loss. The results in the table below shows greater improvement, consistent with Table 2 in the main manuscript.
>
> | Loss | GoPro PSNR | GoPro SSIM |
> | --- | --- | --- |
> | L1 (Original)  | 30.40 | 0.9018 |
> | L1-$a$RGB | 30.85 (+0.45 dB) | 0.9096 (+0.0078) |
>
> ### **[Concluding remark]**
>
> We hope our answers suits your great questions. If any more concern rises, please feel free to ask us at any time. We, again, thank you for your recognition of our contributions, i.e., our analysis and visualizations, the novelty, the versatility and the interpretability of our approach.
>
> ### **[Reference]**
> - [4*] Seungjun Nah et al., “Deep Multi-Scale Convolutional Neural Network for Dynamic Scene Deblurring,” In *CVPR* 2017.
> - [12*] Manuel Fritsche et al., “Frequency Separation for Real-World Super-Resolution,” In *ICCVW* 2019.

---

### Official Review · Reviewer_JdZe · 2023-11-09

**Soundness:** 3 good
**Presentation:** 4 excellent
**Contribution:** 3 good
**Rating:** 6
**Confidence:** 4

**Summary:**

The paper proposes augmented RGB representation to alleviate the issue that per-pixel loss functions defined in the RGB color space tend to produce blurry, unrealistic textures. The proposed aRGB is designed with a nonlinear mixture-of-experts encoder and a linear decoder to meet two requirements. The experiments are conducted on various loss functions across different image restoration tasks for demonstration.

**Strengths:**

The paper analyzes the drawbacks of the per-pixel loss functions in the RGB space, To alleviate the issues of the tendency to producing blurry blurry, unrealistic textures, the paper proposes an aRGB representation to include the local texture for training. The analyses are sound and profound.  Based on the developed encoder and decoder, the method improves the performance of three image restoration tasks using different kinds of loss functions.

**Weaknesses:**

The additional architecture for aRGB representation transmission may introduce more computation consumption during the training phase. The improved performance on image motion deblurring seems to be minimal.

**Questions:**

1. The authors design a nonlinear mixture-of-experts encoder and a linear decoder for aRGB representation. Can this design principle be applied to guide the architecture design of image restoration networks?
2. Is the additional en/decoder equivalent to adding an additional branch for learning? I doubt whether the improved performance is yielded by the additional computation overhead.
3. The widely used dual-domain loss in models, such as MIMOUNet (Cho et al, ICCV'21) and SFNet (Cui et al, ICLR'23), can introduce global information refinement. How does aRGB compare to this loss function? This function does not lead to much computation overhead.
4. Does aRGB lead to extra computation overhead during training and inference?
5. Does the proposed the aRGB architecture rely on the dataset trained on?
6. The reviewer thinks that the performance improvement on GoPro is minimal. For example, only a 0.05 dB PSNR gain is obtained for MPRNet on GoPro. What do the authors think about this?

---

> ### Author Response · Authors · 2023-11-17
>
> First of all, we appreciate your in-depth review and intriguing questions regarding our work. Since we find that the possible weaknesses you have queried are repeating in the Questions section (Questions 4 and 6), we would like to answer each of your questions directly. Your thoughtful questions are invaluable to us for strengthening this work, and we are excited to discuss them with you. Due to the character limit, we split our response into two comments.
>
> ### **[Q1-2: Architecture design]**
>
> > 1. The authors design a nonlinear mixture-of-experts encoder and a linear decoder for aRGB representation. Can this design principle be applied to guide the architecture design of image restoration networks?
> > 2. Is the additional en/decoder equivalent to adding an additional branch for learning? I doubt whether the improved performance is yielded by the additional computation overhead.
>
> - Indeed exploiting experts in architecture may help, as suggested in [3*]. However, we would like to emphasize that our aRGB is not an additional branch for restoration models. Rather, it is similar to how perceptual loss does not count as an architectural augmentation. Thus, our aRGB space does not incur additional computational overhead during the test time, as described in our answer below.
>
> ### **[Q3: Extra computational overhead]**
>
> > 3. The widely used dual-domain loss in models, such as MIMOUNet (Cho et al, ICCV'21) and SFNet (Cui et al, ICLR'23), can introduce global information refinement. How does aRGB compare to this loss function? This function does not lead to much computation overhead.
>
> - We clearly note that our method only applies to the training phase of a restoration model, and therefore **does not cause any additional computational overhead in the test time**.
>
> - For training time, our model does introduce additional computation for the embeddings. **The amount of this overhead is not significant** compared to the image restoration model itself. For example, training RRDB [5*], a widely used super-resolution model, with our $a$RGB loss increases training time about 9%.
>
> - Moreover, the model itself is very lightweight, compared to the massive volume of the current state-of-the-art restoration models. The table below compares the computational burden with other loss functions and restoration models.
>
> |  | # params |
> | --- | --- |
> | $a$RGB encoder* | 5.3M |
> | VGG loss | 20.2M |
> | Adversarial loss (SRGAN discriminator) | 80.2M |
> | ESRGAN | 16.7M |
>
> \*Our $a$RGB decoder has only 0.01M parameters. It is a linear layer. Furthermore, the decoder is not used in the training of image restoration models and therefore its computational cost does not affect the overall performance.
>
> ### **[Q4: Comparison with frequency domain loss]**
>
> > 4. Does aRGB lead to extra computation overhead during training and inference?
>
> - In contrast to the Fourier space-based loss used in MIMO-UNet [6*] and SFNet[7*], our $a$RGB space-based loss has the following three unique characteristics.
>
>     1. Whereas Fourier space loss is used in combination with the RGB space-based counterparts, our $a$RGB loss is designed **as a new alternative for the RGB space**.
>     2. The dual representation in the Fourier space loss is obtained by applying the FFT algorithm to images. This is a constant linear operation, and hence the correlations it models only depend on the coordinate differences, and not the contents of the signal. In contrast, our **$a$RGB encoder is** **nonlinear and adaptive to the input signal**. The metrics in the $a$RGB space expands and contracts based on the images’ local structures as demonstrated in Figure 5d of our manuscript.
>     3. Fourier space loss applies globally within the patch extent, yet its correlation model is fixed. In contrast, our $a$RGB encoder **behaves based on the local image structure**, yet it can **encode more complex structures adaptively**. Figures 5b, 5c and 20 in our manuscript demonstrates its adaptability on the local image structure.
>
> - We design an experiment to further compare the two types of space modeling. Here, a deblurring model [4*] is trained using two types of losses: (1) the Fourier space loss with L1-RGB loss [6*, 7*] and (2) the $a$RGB loss only. Please understand that the time and resources permitted to us is limited, so we are temporarily reporting the intermediate scores. Still, the table clearly exhibits the advantage of our $a$RGB loss over the Fourier loss.
>
> | DeepDeblur / GoPro (100 epochs) | PSNR (dB) | SSIM |
> | --- | --- | --- |
> | Fourier space loss + L1-RGB | 27.0007 | 0.8241 |
> | L1-$a$RGB | 27.7348 **(+0.73 dB)** | 0.8416 **(+0.0175)** |

---

> ### Author Response · Authors · 2023-11-17
>
> (Continued from the last comment)
>
> ### **[Q5: Autoencoder’s dependency on training datasets]**
>
> > 5. Does the proposed the aRGB architecture rely on the dataset trained on?
>
> - **Yes**, as stated in Table 4 and elaborated in Section 4.4 of the main manuscript. However, we tried to **minimize the dependency** by training our $a$RGB autoencoder with a broad set of clean, natural images (DIV2K, Flickr2K, and DIV8K) in order to maximize the versatility of our $a$RGB loss. The results on various restoration tasks in Table 1, 2, and 3 are reported with the same $a$RGB autoencoder model throughout. The result clearly states that that this pre-training stage need not be done more than once. We only have to plug in the pre-trained $a$RGB encoder to the loss function.
>
> ### **[Q6: Significance of the result of image deblurring]**
>
> > 6. The reviewer thinks that the performance improvement on GoPro is minimal. For example, only a 0.05 dB PSNR gain is obtained for MPRNet on GoPro. What do the authors think about this?
>
> - To further justify our approach, we apply our method on another deblurring method [4*], as presented in the table below. Consistent with Table 2 in the main manuscript, our method demonstrates superior performance. The numbers are calculated from the author’s official code repository.
>
> | Loss | GoPro PSNR | GoPro SSIM |
> | --- | --- | --- |
> | L1 (Original) | 30.40 | 0.9018 |
> | L1-$a$RGB | 30.85 **(+0.45 dB)** | 0.9096 **(+0.0078)** |
>
> ### **[Concluding remark]**
>
> We hope our answers clear your concerns. Finally, we deeply thank you for your recognition of our work’s analysis and for your acknowledgement of our contribution to improve performance on various restoration tasks.
>
> ### **[Reference]**
> - [3*] Xiangtao Kong et al., “ClassSR: A General Framework to Accelerate Super-Resolution Networks by Data Characteristic,” In *CVPR* 2021.
> - [4*] Seungjun Nah et al., “Deep Multi-Scale Convolutional Neural Network for Dynamic Scene Deblurring,” In *CVPR* 2017.
> - [5*] Xintao Wang et al., “ESRGAN: Enhanced Super-Resolution Generative Adversarial Networks,” In *ECCVW* 2018.
> - [6*] Sung-Jin Cho et al., “Rethinking Coarse-to-Fine Approach in Single Image Deblurring,” In *ICCV* 2021.
> - [7*] Yuning Cui et al., “Selective Frequency Network for Image Restoration,” In *ICLR* 2023.

---

### Author Response · Authors · 2023-11-17

We express our deepest thanks to all of the reviewers for taking your precious time joining in our discussion on the current drawback of the RGB color representation for image restoration. Your concerns and questions are the most valuable references to us to improve our work and to direct our research in this field.

To our understanding, the reviewers have generally **agreed on our main argument** that *the RGB color space is not the optimal representation for solving image restoration* (**JdZe**, **nF5y**, **cKyU**), the problem that motivated this work. The reviewers have also recognized the **novelty** of our approach (**tXWj**, **cKyU**, **nF5y**) **on designing a loss function for image restoration**. We thank the reviewers for your acknowledgement of the **soundness and comprehensiveness of our analysis** (**JdZe**, **nF5y**, **cKyU**), the **versatility** of our $a$RGB space for loss design (**JdZe**, **tXWj**, **nF5y**, **cKyU**), and its **interpretability** (**nF5y**, **cKyU**). Those you have mentioned were indeed the primary purpose of our work as stated in Section 1 of the manuscript.

Before we get to the individual discussion, we wish to highlight the following point:

To the best of our knowledge, ours is the **first attempt** to **completely remove** the RGB space-based distance losses and yet demonstrating better performance in PSNR metrics. There has been a considerable concern addressing the problem of per-pixel distances in the RGB space, dating back to 2017 [1*, 2*]. However, the use of such loss has been considered necessary, due to the well-known equivalence between the per-pixel $L_{2}$ distance and the peak signal-to-noise ratio metric. **We wanted to take a bold step to remove the per-pixel RGB loss from training a restoration model and see if it is indeed compulsory**. As the paper concludes, **it is not**.

The reviewers have raised several valuable questions. We have tried our best providing answers for every question in your reply section. However, if any concern rises, please feel free to open a new discussion. We are ready to start a discussion with you immediately.

### **[Revision notes]**
1. A typo in Equation 6 is fixed, as the reviewer **nF5y** has pointed out.
2. Tables 1, 2, 3, 4, and 9 are added with dashed lines to make things clearer, in response to the reviewer **tXWj**.
3. Section 2.1 is revised in response to the reviewer **cKyU** to clarify our claim on using larger representation space than the image space.

We appreciate all the reviewers for your constructive reviews.

### **[Reference]**
- [1*] Christian Ledig et al., “Photo-Realistic Single Image Super-Resolution Using a Generative Adversarial Network,” In *CVPR* 2017.
- [2*] Yochai Blau and Tomer Michaeli, “The Perception-Distortion Tradeoff,” In *CVPR* 2018.

---

### Author Response · Authors · 2023-11-22

We deeply appreciate once again for all the reviewer’s valuable time and effort reviewing our work. As multiple reviewers (**JdZe**, **tXWj**) have requested, we have compared our loss with Fourier space loss [6*, 7*] on a deblurring model [4*]. The results we posted in the initial individual comments were the intermediate scores. Here, we report the full score of the experiments, revealing a clear superiority of our $a$RGB space loss. We welcome any discussion with you.

| Loss | GoPro PSNR | GoPro SSIM |
| --- | --- | --- |
| L1 | 30.40 | 0.9018 |
| Fourier space loss + L1 | 30.40 (+0 dB) | 0.9015 (-0.0003) |
| L1-$a$RGB | 30.85 (+0.45 dB) | 0.9096 (+0.0078) |

[4*] Seungjun Nah et al., “Deep Multi-Scale Convolutional Neural Network for Dynamic Scene Deblurring,” In *CVPR* 2017.

[6*] Sung-Jin Cho et al., “Rethinking Coarse-to-Fine Approach in Single Image Deblurring,” In *ICCV* 2021.

[7*] Yuning Cui et al., “Selective Frequency Network for Image Restoration,” In *ICLR* 2023.

---

### Author Response · Authors · 2023-11-22
**Final Remarks**

We would like to express our sincere gratitude once more to all the reviewers and the meta-reviewers for taking their time reviewing our work—amid their busy schedule, especially with the unusual overlapped timeline with the CVPR—and providing fruitful reviews that have definitely improved the paper.

We tried our best to address the raised concerns with the given limited amount of time and resources, but there hasn't been any further dialogues. We hope this implies that our rebuttal has addressed the reviewers' concerns.

We would like to conclude this discussion phase with our final remarks:

- First, we appreciate the reviewers’ acknowledgement of our main problem of *the suboptimality of the RGB color space for image restoration* (**JdZe**, **nF5y**, **cKyU**), the **novelty** of our approach (**tXWj**, **cKyU**, **nF5y**), the **effectiveness of our analysis** (**JdZe**, **nF5y**, **cKyU**), the **versatility** of our $a$RGB space (**JdZe**, **tXWj**, **nF5y**, **cKyU**), and the **interpretability** (**nF5y**, **cKyU**). Those were the main strength we wanted to address in our manuscripts.
- We also would like to note that our main contribution is not the state-of-the-art performance, the lack of which cannot be a reason for rejection, as we carefully refer to the [ICLR guidelines](https://iclr.cc/Conferences/2024/ReviewerGuide#FAQ).
- Rather, we attempt to suggest an alternative and explainable way of training image restoration models with our proposed $a$RGB representation space.
- We ***replace*** the status-quo L1/L2 loss functions over the RGB space with our $a$RGB counterparts, bringing consistent (sometimes marginal, sometimes large) performance gain across various image restoration models and tasks.
- The observed consistent (i.e., no harm) improvements and analyses point to the **potential benefits of replacing the standard RGB space for the per-pixel metrics**.
- We hope our work leads to further research questions and interests in this research avenue.

We, again, appreciate comments from all the reviewers that have indeed helped our work in a constructive way.

---

### Meta-Review · Area_Chair_CBb5 · 2023-12-02

**Metareview:**

The paper proposed a new latent feature space named aRGB for per-pixel distances calculation in loss function design, which benefit the performance in image restoration tasks.

I have thoroughly read all the comments from the reviewers, the authors’ responses, and the paper. The primary concern appears to lie in the novelty of the paper. The key idea of aRGB seems to have similarities to perceptual loss (Johnson et al., 2016) or Fourier loss, even though the authors argue that aRGB is more suitable for accurate reconstruction. In Figure 1(b), it is shown that the aRGB vector can be decomposed into RGB vector and structure vector, where the structure vector represents the latent feature achieved through deep networks. Hence, one might question why not simply use a combination of RGB loss and perceptual loss instead of introducing the aRGB framework?

It is indeed an interesting point that the nullspace of matrix A can be viewed as extra information in the aRGB space, with matrix A serving as the autoencoder's decoder. However, if the additional information is contained in the nullspace of the decoder, then training the autoencoder model may not have a significant impact on this extra information. This makes the main design of the method not very convincing.

**Justification For Why Not Higher Score:**

The level of novelty in the paper is considered insufficient, and some of the claims made are not convincing enough.

**Justification For Why Not Lower Score:**

I have voted to reject.

---

### Decision · Program_Chairs · 2024-01-16

Reject